# Visually-updated hand state estimates modulate the proprioceptive reflex independently of motor task requirements

Sho Ito*, Hiroaki Gomi*

NTT Communication Science Laboratories, Nippon Telegraph and Telephone Co., Kanagawa, Japan

**Abstract** Fast signaling from vision and proprioception to muscle activation plays essential roles in quickly correcting movement. Though many studies have demonstrated modulation of the quick sensorimotor responses as depending on context in each modality, the contribution of multimodal information has not been established. Here, we examined whether state estimates contributing to stretch reflexes are represented solely by proprioceptive information or by multimodal information. Unlike previous studies, we newly found a significant stretch-reflex attenuation by the distortion and elimination of visual-feedback without any change in motor tasks. Furthermore, the stretch-reflex amplitude reduced with increasing elimination durations which would degrade state estimates. By contrast, even though a distortion was introduced in the target-motor-mapping, the stretch reflex was not simultaneously attenuated with visuomotor reflex. Our results therefore indicate that the observed stretch-reflex attenuation is specifically ascribed to uncertainty increase in estimating hand states, suggesting multimodal contributions to the generation of stretch reflexes.

**\*For correspondence:**
sho.ito.fw@hco.ntt.co.jp (SI);
hiroaki.gomi.ga@hco.ntt.co.jp (HG)

## Introduction

Our limb movements often deviate from planned trajectories due to imperfect learning of limb dynamics (*Shadmehr and Mussa-Ivaldi, 1994*), interactions between our body and the environment (*Lackner and Dizio, 1994*; *Lacquaniti and Maioli, 1989*), or the internal noise of neural processing (*Apker and Buneo, 2012*; *van Beers et al., 2004*). To correct movements from these deviations and to stabilize the motor system, feedback control mechanisms are essential. In addition to voluntary correction, a quick implicit sensorimotor process greatly contributes to feedback control.

For instance, a stretch reflex provides the function of feedback control, driven by proprioceptive input. When muscle spindles detect muscle stretch occurring with postural change, the muscle is activated quickly and involuntarily to compensate for the posture or movement error, which is one of the important functions of the stretch reflex (*Marsden et al., 1981*). The stretch reflex occurs earlier (~30–100 ms) than the voluntary response (>~100 ms), yet it also frequently shows goal-directed modulation. Long-latency stretch reflexes (~50–100 ms), which are mainly yielded via transcortical loop (*Evarts and Tanji, 1976*; *Kimura et al., 2006*; *Pruszynski et al., 2011*), in particular exhibit flexible modulation depending on various contexts such as the instability of environments (*Shemmell et al., 2009*), task instructions (*Evarts and Tanji, 1976*; *Hammond et al., 1956*; *Shemmell et al., 2009*), the direction of an up-coming external force field (*Kimura et al., 2006*; *Kimura and Gomi, 2009*), and the spatial properties of visual targets (*Nashed et al., 2012*; *Pruszynski et al., 2008*; *Yang et al., 2011*). Additionally, a visuomotor reflexive response also contributes to quick feedback control. Various studies have demonstrated functional modulation of

quick corrective responses evoked by visual perturbations (cursor shift or background visual motion) (*Abekawa and Gomi, 2010*; *Franklin and Wolpert, 2008*; *Gomi et al., 2006*; *Knill et al., 2011*; *Saijo et al., 2005*).

Properly regulating goal-directed feedback control requires that feedback gain is adjusted with consideration to both context and current body states, including posture and motion of limb. One theory to account for the computational aspect of feedback control is the optimal feedback control (OFC) framework (*Scott, 2004*; *Todorov and Jordan, 2002*). That theory proposed that our brain continuously updates state estimates of the body by combining sensory inflow and internal prediction, in order to provide a state-dependent motor command according to the control policy (*Dimitriou et al., 2013*; *Izawa and Shadmehr, 2008*; *Liu and Todorov, 2007*; *Wagner and Smith, 2008*). Additionally, a number of studies have suggested that the brain integrates multimodal sensory inputs in a statistically optimal manner for perception (*Ernst and Banks, 2002*; *van Beers et al., 2002*) and voluntary motor control (*Ronsse et al., 2009*). However, for reflex control it is still under debate whether feedback responses are calculated from multisensory integration or are only generated by signals within a single modality (*Cluff et al., 2015*; *Oostwoud Wijdenes and Medendorp, 2017*).

As such, the present study examined how altering the properties of visual information, namely by distorting or eliminating visual feedback, affected stretch reflexes. Previous studies of multimodal integration (*Ernst and Banks, 2002*; *van Beers et al., 2002*) predict that these visual changes increase uncertainty of visual information, resulting in more weighting of proprioceptive information. If reflex gain is tuned according to this sensory-weighting, we could expect an enhancement of stretch reflex. On the other hand, these visual changes will reduce the reliability of limb state estimates, due to the combination of visual and proprioceptive signals. If reflexes are regulated according to the reliability of the multimodal state estimates, reflex gain could be reduced to suppress erroneous corrective outputs. Actually, *Izawa and Shadmehr (2008)* have shown that the amplitude of a quick visuomotor response is reduced with increased visual uncertainty. However, this finding was limited to unimodal sensory conditions. It remains unknown whether visual uncertainty also affects proprioceptive reflexes. Therefore, to investigate the effect of multimodal integration on quick sensorimotor control, and to differentiate that effect from goal-dependent modulations, here we focused on the modulation of stretch reflexes by visual information, while keeping hand movements identical across conditions. A part of this data has been preliminarily reported elsewhere (*Ito and Gomi, 2017*; *Ito and Gomi, 2015*).

## Results

### Stretch reflex modulation by visual rotation in visually-guided reaching

In the first experiment, we examined whether distortion of visual feedback alters the stretch reflex gain by applying visual rotations. Participants performed visually-guided wrist flexion, where they moved a cursor representing their hand position toward a visual target (*Figure 1A*). In each experimental block, a particular angle rotation (0˚, 45˚, 90˚, 135˚, or 180˚) was introduced to the movement of the visual cursor (*Figure 1B*). Importantly, locations of the start and the target were also rotated, so that the required hand movements (wrist flexion) were identical across the blocks.

To characterize the effect of visual rotation on the stretch reflex in the flexor (agonist) muscle, a mechanical perturbation (MP) was applied at the wrist joint in the extension direction during wrist flexion movement (*Figure 2A*) for participants in the Agonist group (n = 18). Importantly, variabilities of the endpoint locations of unperturbed trials significantly increased in the large angle rotations (135˚ and 180˚, p<0.05 by the post-hoc comparison after one-way ANOVA with p=3.65 $\times$ $10^{-5}$, $F_{(4, 17)}$=7.67, partial $\eta^2$ = 0.31) as shown in *Figure 2B*, although averaged movement profiles (movement durations, endpoint biases, and peak velocities) of unperturbed trials did not differ significantly from baseline (0˚) by adding visual rotations (*Figure 2—figure supplement 1A - C*). We also confirmed that participants were correctly attending to the cursor movements, by checking correctly shifted movement endpoints in the catch trials (see *Figure 2—figure supplement 1D*) and short reaction times (198 ± 16.1 ms) to the cursor shift in the catch trials of all rotation conditions. *Figure 2C* shows temporal patterns of the wrist flexor muscle activities (rEMG) in three visual rotation conditions of 0˚, 90˚, and 180˚. The rEMGs clearly increased around 30–100 ms after the perturbation onset in all

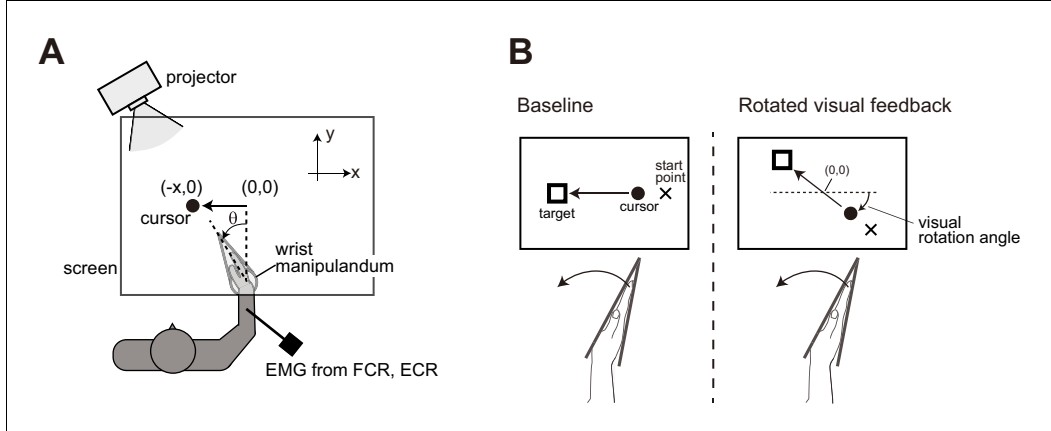

**Figure 1.** Experimental setups. (**A**) Participants' right hand was fixed to a wrist manipulandum, which allowed only flexion and extension movement of wrist joint. Visual feedback was displayed on a horizontally set screen. (**B**) Schematic diagrams of Experiment 1. In the baseline condition, cursor motion was presented along the *x*-axis so that flexion of the wrist joint (increase in *θ*) was transformed to a leftward motion (negative *x*). With the rotated visual feedback condition, locations of the cursor, the starting point, and the goal target were all rotated around the center of work space ([*x*, *y*] = [0, 0]).

three conditions, but peak amplitudes look different. To characterize the stretch reflex modulation, we quantified EMG of short- and long-latency components separately (middle and right panels of *Figure 2D*). Interestingly, the amplitudes of the long-latency stretch reflex were significantly changed by visual rotation angle (one-way ANOVA, p=5.45 $\times$ 10$^{-9}$, $F_{(4, 17)}$=24.77, partial $\eta^2$ = 0.59). Post-hoc analyses indicated that long-latency stretch reflexes in the greater visual rotation ($\geq$90˚) conditions were smaller than that in the baseline (0˚) condition (p<0.05). This means that the gain of the stretch reflex was reduced by introducing a large directional discrepancy between actual hand and visual cursor motions. This reduction is not explained by any changes in the background level of EMG activity (BGA in left panel of *Figure 2D*) which did not differ across conditions (p=0.087, $F_{(4, 17)}$=2.13, partial $\eta^2$ = 0.11). This modulation pattern for the long-latency stretch reflex was not observed for the short-latency stretch reflex (middle panel of *Figure 2D*), although we did find that the short latency components of 90˚ and 135˚ were significantly smaller than the baseline condition (p<0.05 by post-hoc comparison after ANOVA with p=6.07 $\times$ 10$^{-4}$, $F_{(4, 17)}$=5.58, partial $\eta^2$ = 0.25). These results suggest that the reduction of the stretch reflex amplitude more robustly occurred in the long-latency component than the short-latency one.

To examine the effect in the extensor (antagonist) muscle, MP was applied in the flexion direction during the movement (*Figure 3A*) for participants in the Antagonist group (n = 10). As observed in the Agonist group, endpoint variations significantly increased for the large angle rotations (135˚ and 180˚, p<0.05 by post-hoc comparison after ANOVA with p=1.27 $\times$ 10$^{-4}$, $F_{(4, 9)}$=7.78, partial $\eta^2$ = 0.46) from the baseline (*Figure 3B*) while averaged movement profiles of unperturbed trials were not significantly different among visual rotation conditions (*Figure 3—figure supplement 1A - C*). *Figure 3C* shows an example of the temporal patterns of wrist extensor muscle activities (rEMG) in three visual rotation conditions of 0˚, 90˚, and 180˚. As in the Agonist group, significant modulation of the long-latency stretch reflex dependent on the visual rotation angles (one-way ANOVA, p=0.011, $F_{(4, 9)}$=3.83, partial $\eta^2$ = 0.30) was observed (*Figure 3D*). Post-hoc analysis showed significant reduction of the long-latency stretch reflex amplitude from baseline for the greater rotation angles ($\geq$90˚). We did not find a significant difference in the level of background muscle activity (p=0.68, $F_{(4, 9)}$=0.572, partial $\eta^2$ = 0.060), or short-latency stretch reflex amplitude (p=0.30, $F_{(4, 9)}$=1.26, partial $\eta^2$ = 0.12) across the visual rotation angles. In summary, amplitudes of the long-latency stretch reflexes decreased in both the agonist and antagonist muscles, implying that distortions of visual feedback cause reductions in the gain of proprioceptive feedback, regardless of muscle acting direction.

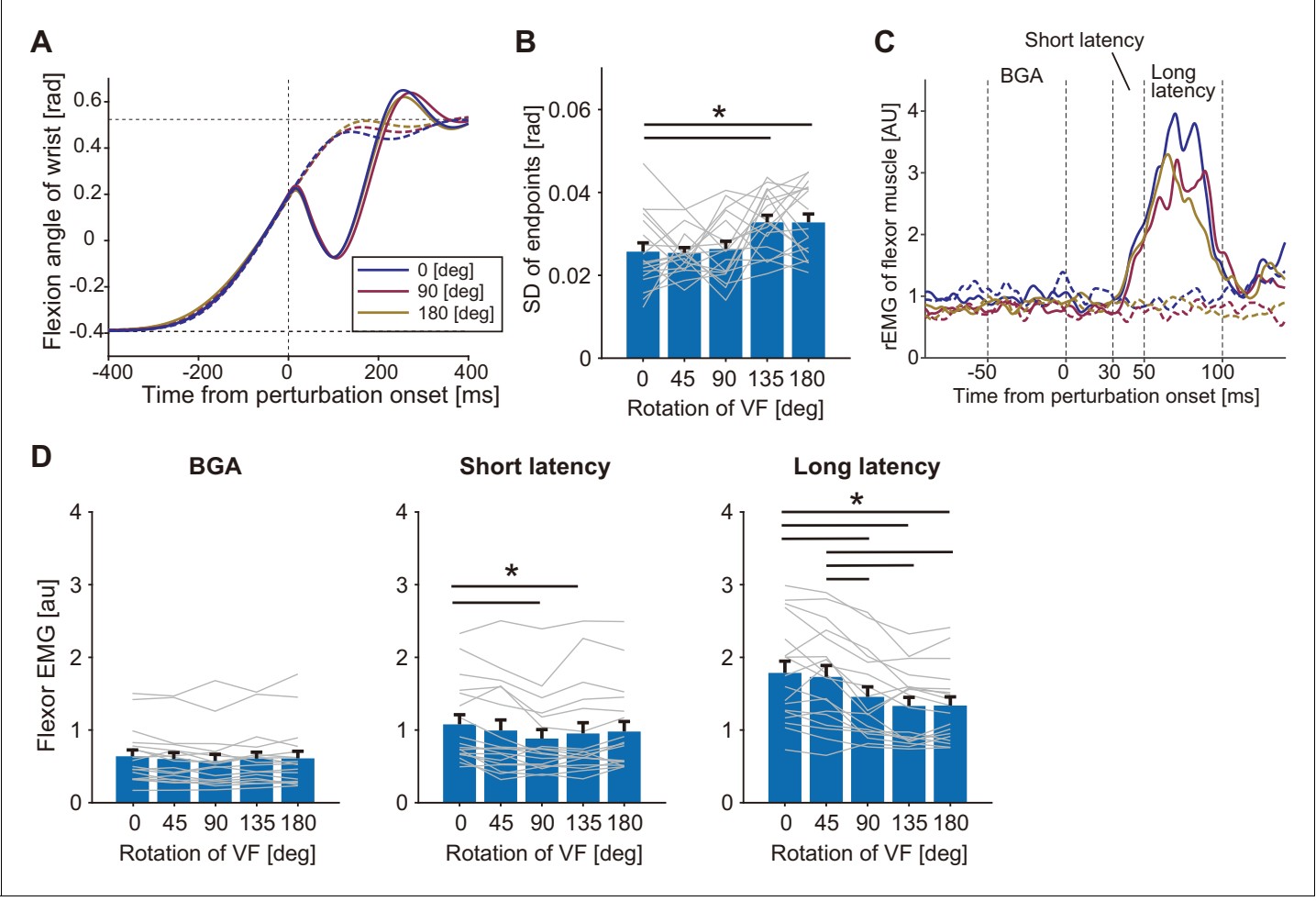

**Figure 2.** Responses to a mechanical perturbation applied during visually guided wrist flexion movements of Agonist group in Experiment 1. (**A**) Temporal profiles of wrist movements with different visual feedback (representative participant). All data were aligned at the position where the mechanical perturbations were applied. Dotted curves show unperturbed trials and solid curves show perturbed trials. Blue, magenta, and yellow curves denote trials with the visual rotation of 0°, 90°, and 180°, respectively. (**B**) Variability of endpoints (standard deviation) of unperturbed trials in each visual rotation condition. (**C**) Rectified and smoothed EMG patterns of wrist flexor of a representative participant. Each solid curve denotes average of perturbed trials and dotted curve denotes average of unperturbed trials (notation of color is same with A). BGA denotes background EMG activity. (**D**) The wrist flexor EMG amplitude in each time window (BGA, Short latency, and Long latency) for each VF rotation (see Materials and methods). Thin gray lines represent data of individual participants. Bar graphs and error bars indicate group mean and standard error across participants. Asterisks indicate significant differences (p<0.05). Additional movement profiles of Agonist group in Experiment 1 are available in figure supplement.
The online version of this article includes the following source data and figure supplement(s) for figure 2:

**Source data 1.** Data of Agonist group in Experiment 1 including quantified muscle activities and movement profiles of each participant.
**Figure supplement 1.** Additional movement profiles of Agonist group in Experiment 1.

## Effect of visual feedback on online modulation of stretch reflex

In the second experiment, we additionally tested the contribution of online visual feedback to the modulation of the stretch reflex. Participants (n = 10) performed the wrist flexion task under normal and mirror-reversed visual feedback (*Figure 4A*) in separate experimental blocks. In half of the trials randomly selected in each block, the visual cursor was eliminated just after the hand movements started as shown in the bottom panels in *Figure 4A*. The averaged kinematic (wrist angle) profiles of mechanically unperturbed trials (dashed curves in *Figure 4B*) did not differ largely by changing the motion direction of visual feedback, or by eliminating the visual cursor. Indeed, a two-way ANOVA comparing peak velocities did not find significant effect of factor for either visual feedback type (p=0.57, $F_{(1, 9)}$=0.35, partial $\eta^2$ = 0.037) or cursor visibility (p=0.68, $F_{(1, 9)}$=0.18, partial $\eta^2$ = 0.020).

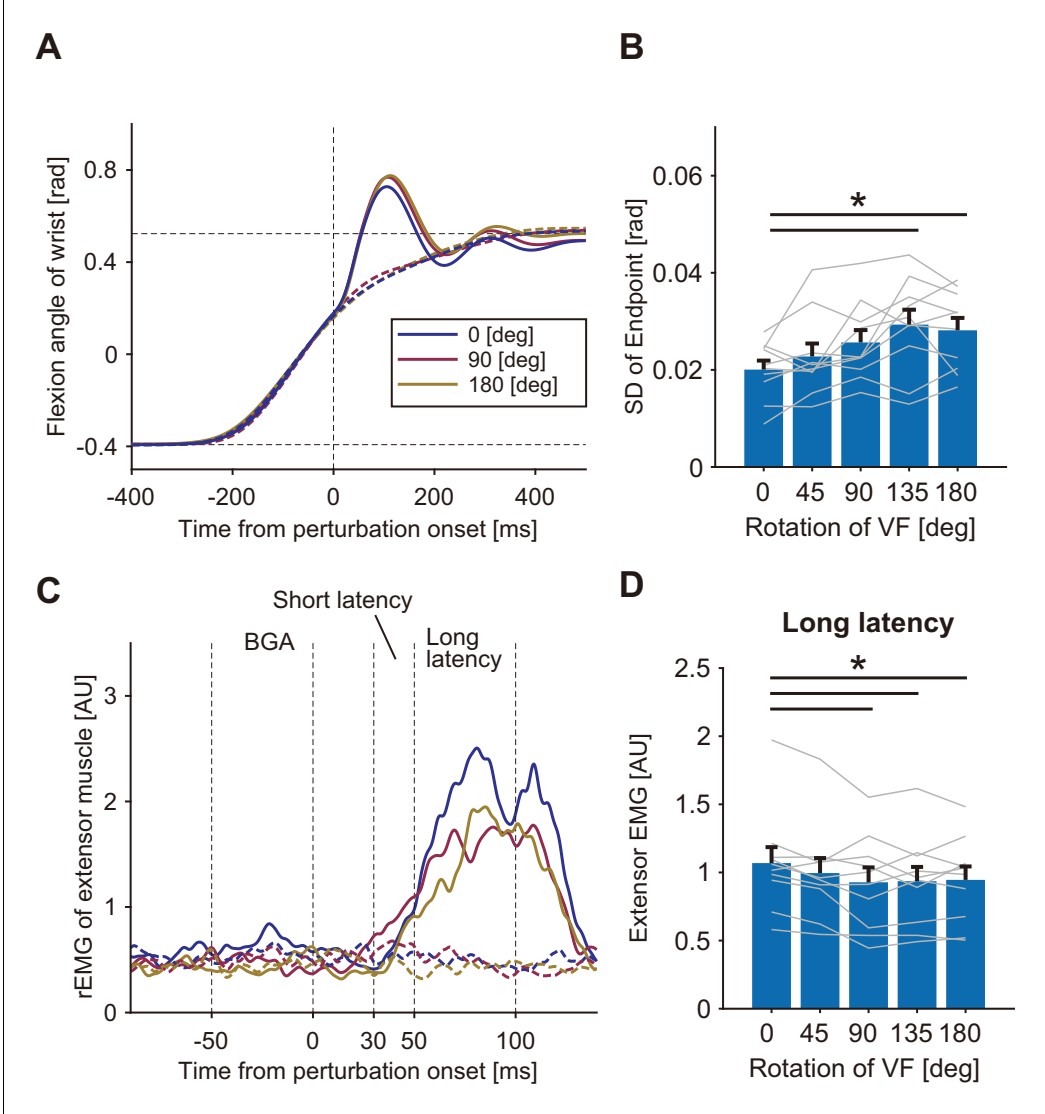

**Figure 3.** Responses to a mechanical perturbation applied during visually guided wrist flexion movements of Antagonist group in Experiment 1. (**A**) Temporal profiles of wrist movements with different visual feedback (representative participant). Dotted curves show unperturbed trials and solid curves show perturbed trials. Blue, magenta, and yellow curves denote visual rotations of 0˚, 90˚, and 180˚, respectively. (**B**) Variability of endpoints (standard deviation) of unperturbed trials in each visual rotation condition. (**C**) Rectified and smoothed EMG patterns of wrist extensor activity for a representative participant. Each solid curve denotes average of perturbed trials and dotted curve denotes average of unperturbed trials (curve color is the same as in A). BGA denotes background EMG activity. (**D**) Amplitude of long-latency stretch reflex of the wrist extensor (means ±SE) for each VF rotation. Lines represent individual data. Asterisk indicates significant differences between the pairs of conditions (p<0.05). Additional movement profiles of Antagonist group in Experiment 1 are available in figure supplement.

The online version of this article includes the following source data and figure supplement(s) for figure 3:

**Source data 1.** Data of Antagonist group in Experiment 1 including quantified muscle activities and movement profiles of each participant.

**Figure supplement 1.** Additional movement profiles of Antagonist group in Experiment 1.

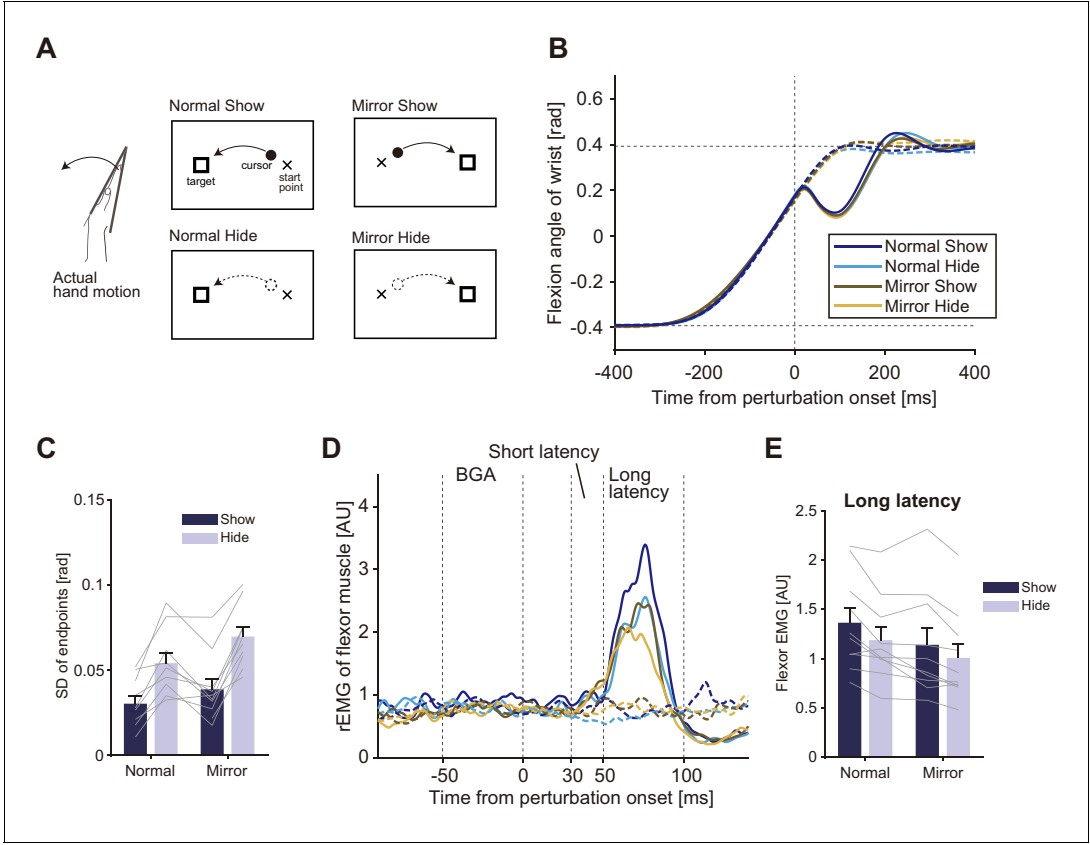

**Figure 4.** Visual feedback effects on stretch reflexes in Experiment 2. (**A**) Experimental conditions and visual feedbacks. Normal and Mirror conditions were inflicted in different blocks while Show and Hide conditions appeared randomly within each block. (**B**) Temporal profiles of wrist movements (representative participant). Data in both perturbed (solid curve) and unperturbed (dotted curve) trials are plotted. (**C**) Standard deviation of movement endpoints (group mean ±SE). Dark blue: data from show-cursor trials; Light blue: data from hide-cursor trials. Grey lines represent individual data. Two-way ANOVA showed significant effects of both visual feedback types. Interaction between the visual feedback type and the cursor appearance was not significant (p=0.10). (**D**) Rectified and smoothed EMG patterns of flexor muscle activity evoked by the mechanical perturbation. Solid curves show perturbed trials and dotted curves show unperturbed trials. Data from a representative participant. BGA denotes background EMG activity. (**E**) Amplitude of long-latency stretch reflexes (group mean ±SE) in Experiment 2. Two-way ANOVA showed significant effects of both visual feedback type (Normal vs Mirror, p=$6.0 \times 10^{-4}$) and cursor appearance (Show vs Hide, p=$1.6 \times 10^{-3}$). Interaction between visual feedback type and cursor appearance was not significant (p=0.41). Additional movement profiles in Experiment 2 are available in figure supplement.

The online version of this article includes the following source data and figure supplement(s) for figure 4:

**Source data 1.** Group data of Experiment 2 including quantified muscle activities and movement profiles.

**Figure supplement 1.** Additional movement profiles in Experiment 2.

As done in the analysis of Experiment 1, we examined the effects of our visual manipulation on the endpoint variability of movements (*Figure 4C*). A two-way ANOVA with factors of visual feedback type and cursor visibility, showed that both mirror-reversed visual feedback (p=$2.8 \times 10^{-3}$, $F_{(1, 9)}$=16.51, partial $\eta^2$ = 0.64) and the elimination of the visual cursor (p=$8.5 \times 10^{-6}$, $F_{(1, 9)}$=81.13, partial $\eta^2$ = 0.90) increased the endpoint variability of movements. But the interaction of the two effects was not significant (p=0.10, $F_{(1, 9)}$=3.26, partial $\eta^2$ = 0.27).

*Figure 4D* shows temporal patterns of EMG responses evoked by the MPs in the four conditions of a representative participant. As shown in these patterns, long-latency components were clearly modulated in different conditions, whose average and inter-subject variabilities are shown in *Figure 4E*. A two-way ANOVA showed a significant main effect of the factor of visual feedback type on the amplitude of the long-latency stretch reflex (Normal vs Mirror, p=$6.0 \times 10^{-4}$, $F_{(1, 9)}$=26.53, partial $\eta^2$ = 0.74), indicating that the long-latency stretch reflex was smaller with the mirror-reversed visual feedback than with the normal visual feedback.

There was also a significant main effect of cursor visibility on the amplitude of the long-latency stretch reflex (Show vs Hide, p=1.6 × 10$^{-3}$, $F_{(1, 9)}$=19.67, partial $\eta^2$ = 0.69), indicating that eliminating the online visual cursor also reduced the long-latency stretch reflex gain. As observed in the endpoint variability, the ANOVA did not find significant interaction between the visual feedback type and the cursor visibility (p=0.41, $F_{(1, 9)}$=0.75, partial $\eta^2$ = 0.077). In other words, the effect of the distorted visual feedback and the effect of the visibility of the online cursor independently contributed to the modulation of the stretch reflex gain. In particular, the significant effect of cursor visibility demonstrates the contribution of visual feedback to the online tuning of the stretch reflex because the cursor was eliminated just after the movement start. Note that, although we found small but significant differences in movement durations and endpoints among conditions (*Figure 4—figure supplement 1*), probably due to the cursor elimination and mirror-reversal, we found significant effect of neither visual feedback type (p=0.61, $F_{(1, 9)}$=0.27, partial $\eta^2$ = 0.029) nor cursor visibility (p=0.091, $F_{(1, 9)}$=3.58, partial $\eta^2$ = 0.28) on the level of background EMG activity. Nor were those visual manipulations associated with significant effects (visual feedback type, p=0.47, $F_{(1, 9)}$=0.57, partial $\eta^2$ = 0.060; cursor visibility, p=0.31, $F_{(1, 9)}$=1.18, partial $\eta^2$ = 0.12) on the movement speed immediately before the MP application in the perturbed trials. Therefore, we can exclude the possibility that the observed modulation of the long-latency stretch reflex was caused by differences in background muscle activity or by differences in movement dynamics across conditions.

In the above analyses of Experiments 1 and 2, we consistently observed that the long-latency reflex component decreased and that the endpoint variance increased in response to altered or eliminated visual feedback. Actually, the negative correlation between the changes in reflex amplitude and in endpoint variance was statistically significant in the Agonist group of Experiment 1 (mean $r$ = −0.46, 1000 bootstrap 95% CI [−0.62, −0.26]) and in Experiment 2 (mean $r$ = −0.73, 1000 bootstrap 95% CI [−0.79, −0.66]). Since the imprecision of endpoints can be ascribed to the increased uncertainty in estimating hand states, the stretch reflex gain reductions could be ascribed to an increase in the uncertainty of the state estimation.

Theoretically, uncertainty of online state estimation gradually increases during movement for a certain duration after elimination of visual feedback (*Wolpert et al., 1995*). Therefore, to further examine our hypothesis, we varied the duration of cursor elimination in Experiment 3 (n = 10). As depicted in *Figure 5A*, in the baseline visual feedback condition (Show), the visual cursor was shown throughout the wrist flexion movement. In the other three cursor-elimination conditions (Short-hide, Middle-hide, and Long-hide. See Materials and methods for details), the visual cursor disappeared when the hand passed a certain location. Although movement durations and speeds were not identical among participants (*Figure 5—figure supplement 1A and C*), actual durations of cursor elimination until the perturbations were appropriately varied among the three hide conditions as shown in *Figure 5B*. As a result, we found a significant increase in endpoint variability of the movements as the duration of cursor elimination became longer (*Figure 5C*, p=1.3 × 10$^{-7}$, $F_{(3, 9)}$=23.00, partial $\eta^2$ = 0.72), which could be due to an increase in the variability of state estimation. Note that we did not find a significant effect on movement duration, endpoints, peak velocity, nor BGA (*Figure 5—figure supplement 1A - D*). Meanwhile, stretch reflex amplitude significantly decreased with longer eliminations of the visual cursor (*Figure 5D*, p=2.6 × 10$^{-3}$, $F_{(3, 9)}$=6.11, partial $\eta^2$ = 0.40). Importantly, correlation between reflex amplitude and endpoint variability was statistically significant (mean $r$ = −0.62, 1000 bootstrap 95% CI [−0.83, −0.33]). This negative relationship is consistent with the aforementioned interpretation, namely that stretch reflex gains are tuned in a manner, which depends on the certainty of the hand state estimation, which in turn utilizes online visual feedback.

## Anti-reaction attenuates quick visuomotor response but not stretch reflex

In the above experiment, we examined the effects of distortion and elimination of visual feedback on the stretch reflex gain. While we hypothesized that uncertainty of multimodal state estimates causes changes in the reflex gain, another possible explanation is that the visual manipulation induces a 'general inhibition' of quick sensorimotor control (*Beritov, 1968*). If this is the case, decreases in response amplitude should not be specific to the visuomotor reflex, but may instead widely affect sensorimotor gains, including the proprioceptive reflex. Therefore, in the fourth experiment we examined whether gain reductions of quick visuomotor responses are always accompanied by gain

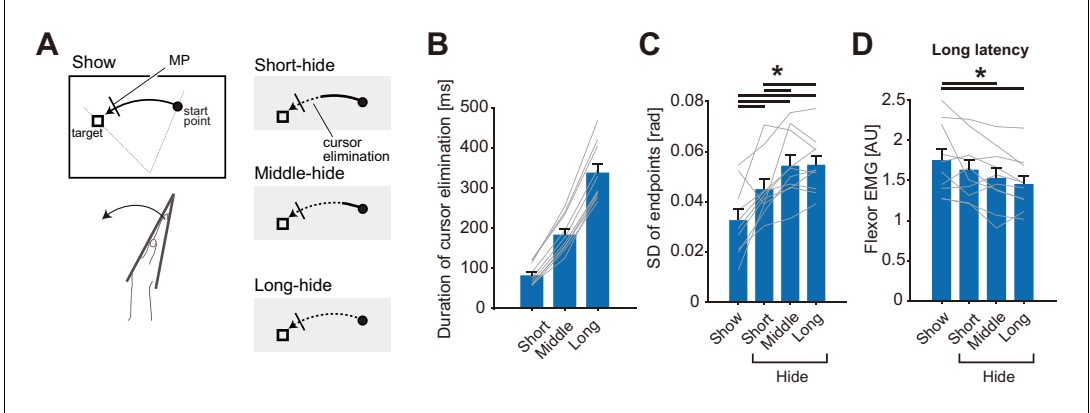

**Figure 5.** Cursor elimination effects on stretch reflexes in Experiment 3. (**A**) Schematic diagram of the experimental task. The visual cursor disappeared (shown as dotted curves in the right panels) after passing one of three different locations in the test conditions, while the cursor was displayed throughout the trial in the baseline condition (Show). Mechanical perturbations were applied at a constant position in all conditions, indicated by a short bar on the movement paths. (**B**) The duration of the elimination of visual cursor before the perturbation onset in each type of trial. (**C**) Standard deviation of movement endpoints in unperturbed trials. (**D**) Amplitude of long-latency stretch reflexes. In panel B, C, and D, lines represent individual data, bar graphs and error bars indicate group mean and standard error, and solid horizontal bars indicate significant differences between corresponding pairs (p<0.05). Additional movement profiles in Experiment 3 are available in figure supplement.

The online version of this article includes the following source data and figure supplement(s) for figure 5:

**Source data 1.** Group data of Experiment 3 including quantified muscle activities, movement profiles, and duration of cursor elimination.
**Figure supplement 1.** Additional movement profiles in Experiment 3.

reductions of the stretch reflex. To separately create similar complexities of hand-state *dependent* and hand-state *independent* visuomotor tasks, we used mirror-reversal and anti-reaction conditions. Participants (n = 8) performed wrist flexion toward the target as in the previous experiments (Baseline in *Figure 6*). They were additionally required to correct their hand movements in response to sudden target jumps (Forward/Backward in *Figure 6*) in order to evaluate reflexive visuomotor responses evoked by target jumps as well as stretch reflex responses. The Pro/Anti-reaction session (*Figure 6A*) tested the effect of the anti-reaction task on these responses, and the Normal/Mirror vision session (*Figure 6B*) tested the effects of mirror-reversal of visual feedback on these responses.

*Figure 7A* shows the temporal patterns of hand accelerations (top panels) and EMG (Flexor: middle panels; Extensor: bottom panels) of a particular participant in the Pro and Anti tasks. The hand acceleration patterns diverged around 200 ms after the target jump (top left panel). As reported in previous studies (*Day and Lyon, 2000*; *Franklin and Wolpert, 2008*), acceleration defined response onset (shown as triangles) was clearly later in the Anti task (top right panel) than in the Pro task (top left panel). Similarly, the EMG response was seen around 120 ms after the target jump in the Pro task, while retardation of flexor and extensor response onset was observed in the Anti task.

We calculated group mean and individual trends of the response latency trigged by target jump for the Pro and Anti tasks (*Figure 7—figure supplement 1*). Since the shortest response latency of EMG among all participants was 120.0 ms for Pro and 180.0 ms for Anti, we calculated the mean EMG activities observed in a time window of 115–175 ms as a measure of the reflexive visuomotor muscle responses (v-EMG) in the analyses below.

*Figure 7B* shows an example of hand accelerations (top panels) and EMG (Flexor: middle panels; Extensor: bottom panels) for target jumps and for no target-jump (baseline) in the Normal and Mirror blocks. In the Normal blocks, flexor muscle activity to forward target jump clearly increased in the time window for characterizing a visuomotor reflex (shaded time range). Similarly, increases in the activity of the extensor muscle for the backward target jump in the Normal blocks were found in this time window. Those flexor and extensor activities produced positive and negative changes in acceleration respectively, which resulted in quick correction of hand motion. In contrast, in the Mirror blocks, both flexor and extensor muscle activities started to change after that time window, according to the direction of the target jump.

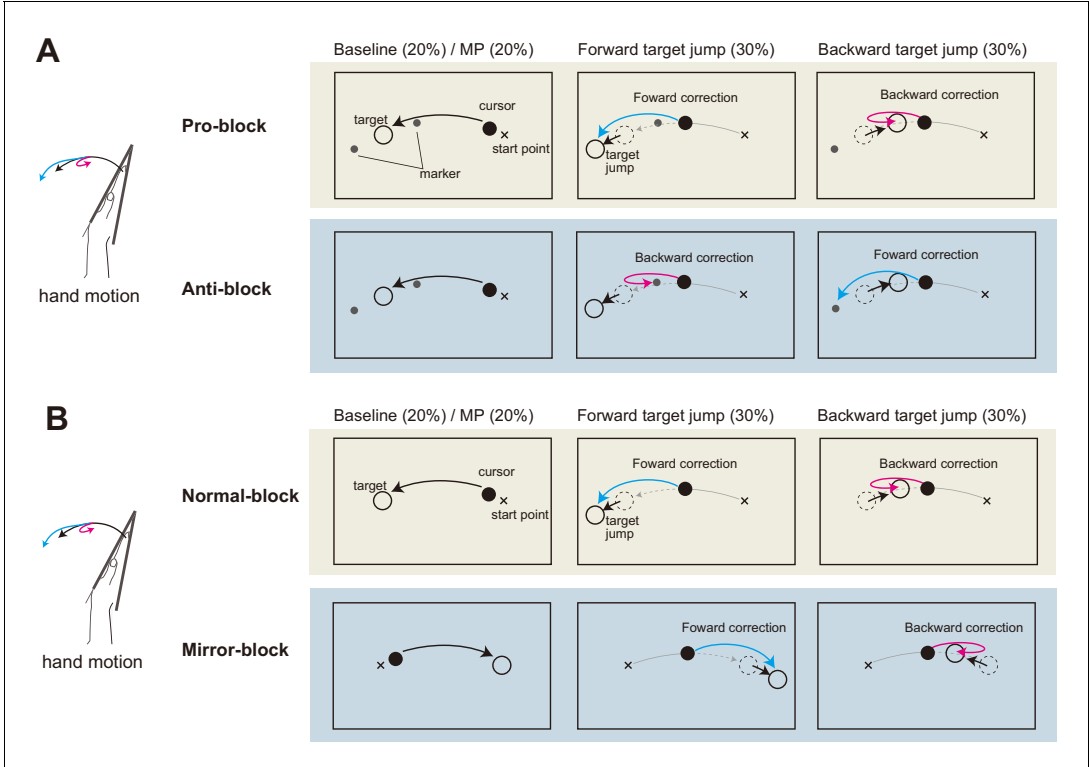

**Figure 6.** Schematic diagram of Experiment 4. (**A**) Pro/Anti-reaction session. Left panels show the task in baseline trial, middle panels show the task in forward target jump trial, and right panels show the task in backward target jump trial. Visual cursor was always shown on the screen. Two small markers (gray dots) were also shown as possible goals in Anti-reaction block. In Anti-reaction block, participant should move their hand in the opposite direction to the target jump (See main text for details). (**B**) Normal/Mirror session. From left to right, panels show baseline, forward target jump, and backward target jump. In Mirror block, hand cursor moved in the opposite direction to the real hand motion.

To examine the condition dependency of the reflexive visuomotor muscle responses (v-EMG) quantitatively, we calculated the differences between v-EMG amplitudes of forward and backward target jump trials (Δv-EMG). *Figure 8A* shows the group mean of Δv-EMG in Normal/Mirror and in Pro/Anti-reaction sessions. Note that movement profiles are shown in *Figure 8—figure supplement 1*. Paired *t*-test showed a significant reduction of flexor Δv-EMG (p=0.013, $t_{(7)}$ = 3.31, *d* = 1.6) in the Mirror blocks compared to that in the Normal blocks. Similarly, flexor Δv-EMG was significantly smaller in the Anti-reaction block than in the Pro-reaction block (p=0.0088, $t_{(7)}$ = 3.60, *d* = 2.3). Compatible results were also obtained for the extensor muscle (*Figure 8—figure supplement 2*). These results indicate that quick visuomotor reaction (reflexive visuomotor response) could not be generated (i.e., was almost completely eliminated) in Mirror and Anti-reaction blocks.

In contrast, we did not find any significant difference in long-latency stretch reflexes between Pro- and Anti-reaction blocks (Left panel of *Figure 8B*, p=0.78, $t_{(7)}$ = −0.29, *d* = 0.020) while the stretch reflex was smaller in the Mirror blocks than in the Normal blocks (Right panel in *Figure 8B*, p=0.0062, $t_{(7)}$ = 3.86, *d* = 0.66). These results therefore indicate that reductions of visuomotor reflexes are not always accompanied by gain reductions of the stretch reflex.

The above contrast between modulations of reflexive visuomotor and proprioceptive-motor responses does not support the hypothesis that gain reductions of the stretch reflex are due to a *general* inhibition of reflexive motor responses, operating across feedback modalities. Instead, it specifically supports the hypothesis that the stretch reflex gain reduction observed in this study can be ascribed to the uncertainty of state estimation caused by distorted or eliminated visual feedback.

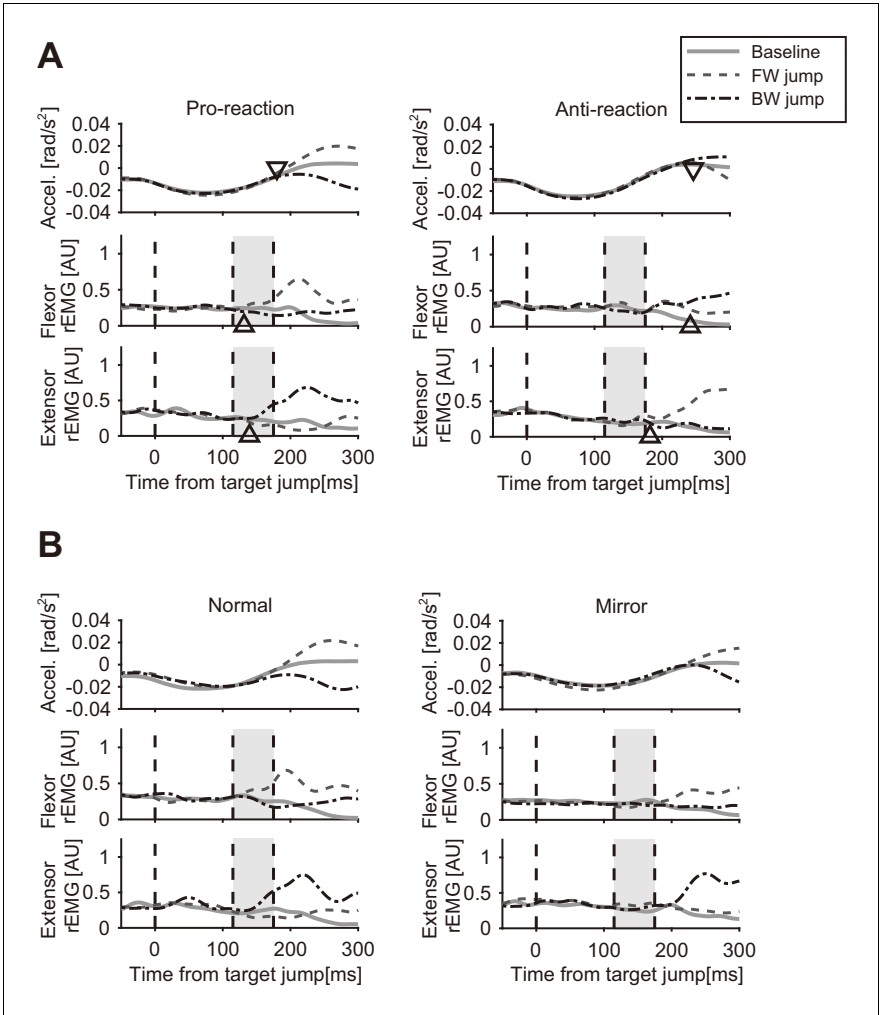

**Figure 7.** Temporal patterns of hand acceleration, Flexor rEMG, and Extensor rEMG. (**A**) Pro/Anti-reaction session (representative participant). All data were aligned to the onset timing of the target jump. Gray solid line: baseline; Dashed line: Forward target jump; Dash-dot line: Backward target jump. Open triangle denotes response latency estimated from each type of data (see Materials and methods). Grey shaded areas show the time window for quantifying quick visuomotor response defined from the reaction latencies of all participants (115–175 ms, see Results). Response latencies for target jump in Pro/Anti-reaction sessions are available in figure supplement. (**B**) Normal/Mirror session (representative participant). Regardless of visual feedback type (Normal/Mirror), forward target jumps required more flexing correction then backward jumps required extending correction. All notations are same as in A. Group data of the response latency are available in figure supplement.

The online version of this article includes the following figure supplement(s) for figure 7:

**Figure supplement 1.** Response latencies for target jump in Pro/Anti-reaction sessions.

## Discussion

The present study assessed the contribution of visual feedback to the gain modulation of the stretch reflexes. The first and second experiments showed that distorted visual hand cursor feedback and elimination of the cursor during movement, in both case significantly reduced the amplitude of long-latency stretch reflexes. The third experiment examined the temporal development of the reflex gain reduction after the cursor elimination, and the forth experiment demonstrated a dissociation between the gain modulations of the quick proprioceptive-motor and visuomotor responses by using different visual feedback conditions. Here we discuss a new aspect of the regulation mechanisms of the stretch reflex.

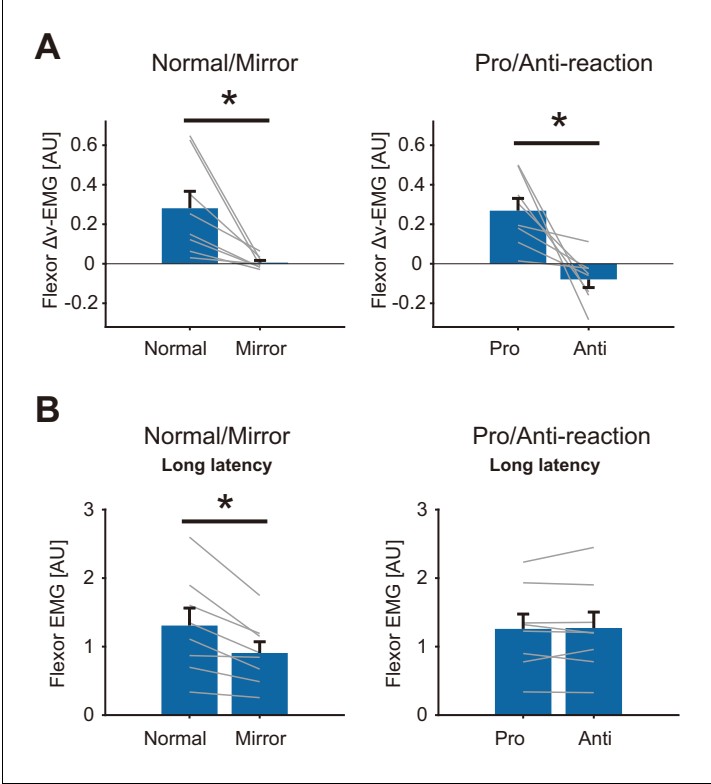

**Figure 8.** Reflexive muscle reactions (Flexor) to target jump and to mechanical perturbation in Normal/Mirror and in Pro/Anti-reaction sessions of Experiment 4. (**A**) Mean flexor Δv-EMG amplitudes, obtained by (v-EMG for forward correction) − v-EMG for backward correction). Δv-EMG in Pro-reaction condition was greater than that in Anti-reaction condition. Δv-EMG in Normal visual feedback condition was greater than that in Mirror condition. (**B**) Mean flexor long-latency stretch reflex amplitudes induced by mechanical perturbation. While a difference in stretch reflex was not found between Pro- and Anti-reaction conditions, the stretch reflex in the Normal condition was greater than that in the Mirror condition. In all graphs, error bars indicate standard error, and thin gray lines represent individual data. Asterisks indicate significant differences. Additional movement profiles in Experiment 4 are available in *Figure 8—figure supplement 1*, and reflexive muscle reactions (Extensor) to target jump in Pro/Anti-reaction and in Normal/Mirror sessions are available in *Figure 8—figure supplement 2*.

The online version of this article includes the following source data and figure supplement(s) for figure 8:

**Source data 1.** Group data of Experiment 4 including long-latency stretch reflex, Δv-EMG, response latencies, and movement profiles.
**Figure supplement 1.** Additional movement profiles in Experiment 4.
**Figure supplement 2.** Reflexive muscle reactions (Extensor) to target jump in Pro/Anti-reaction and in Normal/Mirror sessions of Experiment 4.

## Visual stimulus effects on motor correction

By using distorted visual feedback (e.g., visual rotation and mirror reversal), several studies have investigated the contribution of the visuomotor transformations to the process of generating a visually evoked corrective response. However, it was unclear whether visual information affects only the visuomotor reflex. The present study demonstrates that the distortion of visual feedback can indeed reduce the gain of the stretch reflex, indicating a contribution of visual information to the tuning of the proprioceptive reflex as well.

Under conditions of mirror-reversed visual feedback, an early response to sudden cursor shifts is initially generated toward the incorrect direction (*Gritsenko and Kalaska, 2010*; *Kasuga et al., 2015*; *Telgen et al., 2014*), but participants can learn to suppress this incorrect response after extensive training. In addition, participants can attenuate the early reaching correction evoked by a target jump in anti-target jump tasks (*Gritsenko and Kalaska, 2010*). From those observations, one could imagine that suppressions of visuomotor reflexes are also accompanied by suppression of the

stretch reflex, due to a possible general inhibition of reflexive sensorimotor processing (*Beritov, 1968*). Results of Experiment 4, however, do not accord with this idea. The amplitude of the stretch reflex under the anti-target jump condition was comparable to that under the pro-target jump condition, while it decreased as a result of the mirror-reversal of visual feedback. This discrepancy clearly indicates that the suppression of the visuomotor reflex does not necessarily lead to attenuation of the stretch reflex. Instead, our data imply that stretch reflex modulation occurs when distortion of visual feedback affects the state estimation of the motor system, as in the cases of the rotation and mirror-reversed conditions. Importantly, the anti-target jump task requires a change of the mapping from the visual target to the motor command, which is irrelevant to hand position representation. Therefore, this task would not affect the hand state estimation process of the motor system.

Crucially, we also found that altering visual-feedback increased the endpoint variability of movements in the current study. Many visual rotation studies (e.g., *Cressman and Henriques, 2009*; *Mazzoni and Krakauer, 2006*; *Saijo and Gomi, 2012*; *Saijo and Gomi, 2010*) have focused on the bias effects (endpoint shift) during two dimensional movements, while cursor elimination is well known to increase the variability of endpoints (*Elliott, 1988*; *Proteau, 1992*) and the uncertainty of state estimation (*Wolpert et al., 1995*). In contrast, we did not observe any biasing of movement endpoints in our rotated conditions, relative to the baseline condition. This may be because our one dimensional movement was much simpler than the two dimensional movements used in previous studies. In general, large angles of visual rotation, as well as mirror-reversals, increase reaction times (e.g., *Fernandez-Ruiz et al., 2011*; *Saijo and Gomi, 2010*; *Telgen et al., 2014*). This suggests that these visuomotor transformations are too complex for the nervous system to easily reconcile the mismatch between visual and proprioceptive information. As a result, movement trajectories are expected to become variant by visual feedback distortion, even after some amount of learning. This has been shown in a previous study (*Buch et al., 2003*). These observations therefore imply that distorted visual feedback could deteriorate the integration of visual and proprioceptive signals, in turn limiting the ability of the system to represent hand states, resulting in large variances of movement endpoints for large visual rotation conditions.

## Stretch reflex modulation

Previous studies demonstrated context-dependent modulations of the long-latency stretch reflex according to task instruction, action intention, postural state, and external force field, as mentioned in the Introduction. Meanwhile, few studies (*Crevecoeur et al., 2016*; *Mutha et al., 2008*; *Yang et al., 2011*) have investigated how visual information contributes to the modulation of stretch reflexes. *Mutha et al. (2008)* measured stretch reflexes when goal location changed at the onset of reaching movements, and found increases in the long-latency stretch reflexes toward a new goal location followed by a late (voluntary) increase in muscle activity. This suggests a goal-directed modulation of stretch reflexes by visual information. However, that study did not examine whether visual feedback is utilized in representing limb state for the tuning of stretch reflexes.

To examine this point, we manipulated the properties of visual feedback (visual rotation and removing online cursor) without altering motor-task goals. As shown in the results of Experiments 1 and 2, changes in visual feedback reduced the amplitude of long-latency stretch reflexes without any changes in baseline movements or muscle activity (*Figures 2*, *3* and *4*). These results suggest a new role of visual information, distinct from that used for goal-directed modulation of stretch reflexes. In contrast to our results, *Crevecoeur et al. (2016)* reported less contribution of visual feedback to stretch reflexes because they did not find any modulation in the corrective response to mechanical perturbations by the elimination of a visual cursor. This dissociation in reflex modulation in the current and previous studies could be due to task differences. In particular, the previous study employed a relatively static regulation task, where participants were required to recover a hand position to the position just before the mechanical perturbation. Since participants could detect the static postures before and after the perturbation using their proprioceptive information, they potentially completed the task by relying less on visual feedback. Meanwhile, in our experimental paradigm, participants should rely on visual information because they had to reach visual targets shown in the external workspace rather than recovering a posture. This relatively high importance of visual feedback present in our experimental task has brought to light the significant impact of vision on the stretch reflex.

## Uncertainty in state estimates affects feedback gain

The stretch reflex modulation shown by our visual manipulation clearly supports the contribution of visual information of hand states to stretch reflex gain tuning. However, the observed reduction of stretch reflex amplitude cannot alone be explained by a simple sensory re-weighting between vision and proprioception. Since the theory of optimal sensory integration predicts a heavier weight on proprioception with uncertainty in visual information (*Ernst and Banks, 2002*; *van Beers et al., 2002*), an increase in the stretch reflex gain should be expected. However, we observed a decrease in stretch reflex gain.

To avoid possible inappropriate or harmful motor output due to unreliable state estimates, feedback gain should be reduced as mentioned in the Introduction. Indeed, several experimental studies have demonstrated that uncertainty of state estimate impacts feedback gain in online motor control. For example, *Izawa and Shadmehr (2008)* systematically manipulated the ambiguity of target location and showed that increased uncertainty in the visually estimated state causes reduction in the gain of visuomotor correction. Other studies have shown a decrease in the visuomotor reflex immediately after a dynamic update of visual information due to a target jump (*Dimitriou et al., 2013*) or saccade (*Abekawa and Gomi, 2015*), suggesting that uncertainty caused by a visual update decreases the reflex gain. Likewise, the present experimental manipulation of visual feedback could have made visual information unreliable (*Saijo and Gomi, 2012*; *Wei and Körding, 2010*). Theoretically, with the low reliability of visual feedback, limb states obtained by integrating visual and proprioceptive information also becomes less reliable, even though the optimal integration process maintains as much reliability as possible. In such situations, lower feedback gain would be preferable because larger feedback gain increases the risk of enlarging movement error by generating a large incorrect response due to erroneous state estimation. Consequently, if we assume the generation of stretch reflexes involves integration of multiple sensory sources, it is reasonable to expect that uncertainty in visual feedback will reduce stretch reflex amplitude, compared to situations where visual information is fully available.

Interestingly, *Franklin et al. (2012)* showed increases rather than decreases in the visuomotor reflex during force field adaptation, suggesting that uncertainty of limb or environment dynamics causes up-regulation of reflex gain. The optimal direction of the gain modulation may therefore depend on where the uncertainty originates. In situations with uncertainty in body dynamics or an unpredictable environment, higher gain would be required for quick error correction caused by the uncertainty (*Franklin and Wolpert, 2008*; *Franklin et al., 2012*; *Shemmell et al., 2009*). However, if, as in our study, uncertainty exists in sensory information, smaller gain is desirable (*Izawa and Shadmehr, 2008*; *Körding and Wolpert, 2004*) because it is not beneficial to generate a possibly erroneous response based on unreliable state estimates.

Next, we will consider how the reduction of stretch reflex gain can be explained by integrating sensory signals from different modalities. Even if we assume the different processing delays of proprioceptive and visual signals (*Crevecoeur et al., 2016*), decreases in the stretch reflex would not be predicted. Instead, we need to consider the possibility that feedback gain is regulated according to the reliability of limb states estimated via multimodal integration. For instance, according to the optimal control model, one possibility explaining reflex gain reduction is that the optimal policy calculating optimal feedback gain is modified to evaluate the uncertainty of the state representation. In such a case, the short latency stretch reflex, which might not be involved in high level state estimation, could be altered by top-down regulation, as well as the long latency stretch reflex. The simultaneous reductions in short-latency and long-latency components in Experiment 1 (Agonist group, 90° and 135° conditions) may be partly explained by this hypothesis. Another possibility is that the uncertainty increase in estimated states induces a reduction of the filter gain in the state estimation process, which could lead to a tracking delay in state estimation, resulting in a reduction of the corrective command to a transient perturbation. This possibility was examined by a model for visuomotor response (*Izawa and Shadmehr, 2008*). By considering the current findings regarding the influence of state uncertainty on the regulation of the proprioceptive-motor response shown in the current study, it seems possible that future studies will be able to extend the computational model to include multimodal integration for reaching movement. In both possibilities, to realize the predefined speed of movement we may need to assume that at least a part of the motor command is

generated in a feedforward fashion, as has been examined previously (*Bastian, 2006*; *Kawato, 1999*; *Sabes, 2000*; *Saijo and Gomi, 2010*; *Wolpert et al., 1998*).

## Visual effect on stretch reflexes in multiple stages of the motor control hierarchy

Additive modulations of the stretch reflex by visual discrepancy and elimination observed in Experiment 2 (*Figure 4E*) suggest that the stretch reflex is regulated by multiple factors in the motor-control hierarchy. One possible account is that the gain of the stretch reflex is regulated at two different stages of sensorimotor processing: motor planning and online-control. Many previous studies demonstrated that feedback gain is set before starting movement, depending on various contexts (*Ahmadi-Pajouh et al., 2012*; *Evarts and Tanji, 1976*). This planned reflex control would occur not only in a preset manner (*Bonnard et al., 2004*) but also in a time-varying manner (*Kimura and Gomi, 2009*). In addition to such planned reflex control, the reflex gain could also be updated during movement according to online sensory information (*Mutha et al., 2008*). Therefore, independent and additive gain control mechanisms could be assumed to exist in the neural processing of both movement planning and online-control.

Another plausible account of the additive effect in reflex gain change is that it is due to a gradual increase in uncertainty by cursor elimination combined with uncertainty caused by visual feedback distortion. As shown in Experiment 3, stretch reflex gain decreased with movement variability (*Figure 5C and D*), possibly related to the increase in uncertainty of state estimation over time (*Wolpert et al., 1995*). If the cursor elimination gradually increases the uncertainty of the state estimates under both of the normal and distorted visual feedback conditions, this account would predict similar temporal decays of reflex gain during the movements in those two conditions, which is consistent with the observed additive effect of visual feedback distortion and elimination on the reflex gain reduction.

In both of possible accounts above, the uncertainty of state estimation is an essential factor in stretch-reflex gain control. The present study demonstrated for the first time a contribution of visual feedback of the limbs to the modulation of the stretch reflexes independent of motor task. The results suggest that multimodal integration is involved in state estimation and that the uncertainty of this state estimation underlies the functional tuning of automatic feedback control. Understanding this hidden linkage between state uncertainty and reflex modulation is extremely important since regulation of proprioceptive-motor loop is essential in a wide range of motor controls.

## Materials and methods

### Participants

In total, 48 healthy volunteers (16 males, 32 females; age range 20–49, average 32 ± 9.1) participated in a series of experiments. Out of all the participants, four were tested in three of the experiments, seven were tested in two of the experiments, and the rests were tested in one experiment. Since seven participants out of 35 in Experiment1 could not satisfy a prerequisite of the task performance (See Data collection and analysis for details of the prerequisite), we excluded them from the analysis. All of the volunteers were right-handed. All gave written informed consent to participate in the experiments. All of the experimental protocols were approved (H28-011 and H31-007) by NTT Communication Science Laboratories Ethics Committee.

### Apparatus

Participants sat in front of a horizontally set screen with their right hand tightly fixed to a custom-made manipulandum (maximum torque of 7.0 Nm) for wrist joints as shown in *Figure 1*. The right forearms of the participants were held on an armrest and tightly fastened by a belt. Hand movement was restricted to one degree-of-freedom movement (flexion and extension of wrist joint in a horizontal plane). The manipulandum was controlled by a digital signal processor (iBIS DSP7101A, MTT Co., Tokyo, Japan) at a 2000 Hz control frequency. Vision of the participant's actual hand was occluded by a screen placed over the hand. Feedback of hand movement was provided by a visual cursor that was displayed on the screen via a projector (K335, Aser Inc, New Taipei City, Taiwan). Visual cursor position was updated as wrist angle changed. The instructed movement duration was 750 ms in all

experiments, which was communicated to participants via beeping sounds presented at start of the trial and at the point of expected movement completion. To control movement speed, if the hand reached the middle point of the flexion earlier than 375 ms after trial onset, the trial was discarded and immediately restarted. In addition, if the hand movement stopped later than 750 ms after the start, an alert message was displayed on the screen. Visual and auditory stimuli were controlled via custom made programs using MATLAB (Mathworks Inc, Natick, MA, USA) and Cogent graphics toolbox (developed by John Romaya at the LON at the Wellcome Department of Imaging Neuroscience). A photodiode was placed on the corner of the screen to detect the timings of stimulus changes, and its signal was recorded simultaneously with the other signals.

## Experiment 1

The purpose of the experiment was to examine the effect of distortion of visual feedback on the amplitude of the stretch reflex. In the baseline condition, the visual cursor moved along the x-axis (left-right direction) according to the change in wrist angle, where displacement of the cursor x was calculated as $x = -\alpha\theta$. Here, $\theta$ was a wrist flexion angle from straight hand posture, and $\alpha$ (0.44 cm/deg) was the visual feedback gain. The origin of the coordinates was 25 cm away from the position on the screen above the rotation center of the manipulandum.

Participants were required to move a visual cursor from a starting point (x = −10.0 cm, $\theta$ = −22.5°) toward a visual target (x = −10.0 cm, $\theta$ = 30°) by flexing their wrists. We prepared five types of experimental blocks. In each block, visual rotation of the cursor movement was introduced in only one of the following directions: 0°, 45°, 90°, 135°, or 180°, around the origin of the coordinates (*Figure 1B*). Here, the visual locations of start and goal markers were also rotated to the same degree as the cursor rotations. In other words, required hand movement remained constant, while the degree of visual rotation changed in a block-dependent manner. The visual cursor was present throughout the trial, during both initial reaching movement (wrist flexion) and subsequent return movement (wrist extension).

In order to evoke stretch reflexes, a mechanical perturbation (half sine-wave torque pattern with 50 ms duration, 2.0 Nm peak amplitude) was applied randomly in 25% of trials. The perturbation was applied in the direction of wrist extension to evoke stretch reflexes from flexor muscles for the halfone group of the participants (Agonist group, n = 18), and was applied in the opposite direction to evoke stretch reflexes from extensor muscles for the others (Antagonist group, n = 10). The perturbation was initiated when the hand passed a predefined trigger position ($\theta$ = 3.75°). Note that actual hand positions when the perturbation was applied were slightly more flexed (mean ± SD across conditions: 11.03°±0.14°) than the trigger position due to a system delay in our experimental setup. There was no statistical difference between the wrist angles at which perturbations were applied in all the experimental blocks (Agonist group, p=0.34, $F_{(4, 17)}$=1.16, partial $\eta^2$ = 0.064; Antagonist group, p=0.15, $F_{(4, 9)}$=1.78, partial $\eta^2$ = 0.17). To keep the participants from executing hand movement without visual feedback, catch trials were included, in which the cursor position abruptly shifted (±12°) at the midpoint of the reaching movement. Participants were asked to compensate for the shift by bringing the cursor to the target as quickly as possible when the shift was perceived.

The experimental block consisted of 20 trials (five for no visual perturbation and no mechanical perturbation (N), five for mechanical perturbation (MP), five for forward cursor shift, and five for backward cursor shift. All trials were ordered randomly). All of the five experimental blocks with different visual rotation angles were ordered pseudo-randomly and repeated five times.

## Experiment 2

In this experiment, we examined the contribution of online visual feedback to the tuning of stretch reflexes. For this purpose, we hid the visual cursor in half the trials, in addition to a directional change of visual feedback.

As in Experiment 1, participants (n = 10) were asked to make wrist flexion movements. We prepared two types of experimental blocks where the direction of cursor movement differed (*Figure 4A*). In one type of block, visual feedback moved in the same direction as actual hand motion, like with the 0° condition in Experiment 1 (Normal block). In another type of block, the

direction of cursor movement was mirror-reversed relative to the actual hand movement (Mirror condition), which was nearly equal to the 180° condition in Experiment 1.

In both conditions, hand posture at the start was identical ($\theta = -22.5°$), and three target postures (Standard: Std, $\theta = 22.5°$; Near, $\theta = 16.5°$; and Far, $\theta = 28.5°$) were randomly applied, to prevent participants from performing the movement without visual information. In the Mirror condition, start and target locations were also mirror-reversed so that required hand movements were identical to those in the Normal condition. For half of the trials for each target, the hand cursor was shown throughout the trial (Show trials). In the other half, the hand cursor was hidden just after the detection of movement onset ($\dot{\theta} > 30$ deg/s) (Hide trials), and reappeared after the stop ($\dot{\theta} < 30$ deg/s). In the Show and Hide trials for the Std target, the mechanical perturbation (MP) extended the wrist when hand passed a trigger position ($\theta = 0°$) to evoke the stretch reflexes in 40% of trials. In the other 60% of trials for the Std target, MP was not applied (N). Each experimental block consisted of 40 trials (12 Std w/o MP, 8 Std with MP, 10 Near, and 10 Far), half of which were Hide trials. We repeated a pair of Normal and Mirror blocks (order was randomized) six times. The order of all trials in each block was randomly shuffled. Conditions are summarized in *Table 1*.

## Experiment 3

This experiment was designed to extend the earlier examination of the contribution of online visual feedback to stretch reflex amplitude, by manipulating the duration of cursor elimination. As in the other experiments, participants (n = 10) performed wrist flexion task from a starting point ($\theta = -22.5°$) to a visual target (standard target: Std, $\theta = 45°$), as shown in *Figure 5A*. In a baseline visual feedback condition (Show), the visual cursor was displayed at a fingertip position throughout the flexion movement. In the other three cursor-elimination conditions (Short-hide, Middle-hide, and Long-hide), the visual cursor disappeared when hand passed a certain location for each condition (+33.8° flexion for Short-hide, +16.9° flexion for Middle-hide, and +1.0° flexion from the starting point for Long-hide) as depicted in *Figure 5A* and reappeared after the movement stopped to show endpoint position. In randomly selected trials (40% of each condition), the mechanical perturbation (MP) suddenly extended the wrist when the hand reached a constant trigger position (+50.6° flexion from the starting point) to evoke the stretch reflex. In the remaining trials (60%) with a Std target, the mechanical perturbation was not applied (N).

To prevent participants from memorizing target location, we also required them to reach two additional targets (Near target, $\theta = 39°$ and Far target, $\theta = 51°$) with Show and Long-hide conditions,

**Table 1.** Trial conditions in Experiment 2.

| Block (Direction) | Target | Cursor | Perturbation | # of trials |
|---|---|---|---|---|
| Normal | Std | Show | MP | 4 |
| | | | N | 6 |
| | | Hide | MP | 4 |
| | | | N | 6 |
| | Near | Show | N | 5 |
| | | Hide | N | 5 |
| | Far | Show | N | 5 |
| | | Hide | N | 5 |
| Mirror | Std | Show | MP | 4 |
| | | | N | 6 |
| | | Hide | MP | 4 |
| | | | N | 6 |
| | Near | Show | N | 5 |
| | | Hide | N | 5 |
| | Far | Show | N | 5 |
| | | Hide | N | 5 |

in which the mechanical perturbation was not applied (N). One experimental block consisted of 80 trials (10 trials each for Show and the three hide conditions with Std target, 10 trials each for Show and Long-hide with Near target, and 10 trials each for Show and Long-hide with Far target) and was repeated six times. Trial conditions in each block are summarized in *Table 2*.

## Experiment 4

The purpose of this experiment is to examine whether gain modulations of visual and proprioceptive reflexes occur jointly or independently. The experiment consisted of two types of sessions: Pro/Anti-reaction session and Normal/Mirror vision session. The latter session was included again in this experiment so as to compare the reflex modulations of those conditions within the same participants. Each participant (n = 8) performed both of the sessions, and session order was counter-balanced across participants. In each session, we measured both the stretch reflex evoked by mechanical perturbations and the visuomotor responses elicited by target jumps.

The session involving Pro/Anti-reaction blocks was designed to characterize the effect of the anti-reaction condition on visuomotor and proprioceptive-motor responses. Throughout this session, the moving direction of the visual hand cursor was consistent with that of the actual hand. Participants were instructed to flex their wrists from the start position ($\theta = -22.5°$) to the standard target ($\theta = 22.5°$) with the assistance of a visual cursor. As shown in the left panels of *Figure 6A*, small markers were shown as possible positions of the jumped target. In Pro-reaction blocks (Pro), participants were also asked to correct their hand movement as quickly as possible when the target jumped forward or backward (±12°, 30% trials for each, as drawn in the middle and right panels of *Figure 6A*). The target jump occurred when the hand passed a position of $\theta = 0°$. In Anti-reaction blocks (Anti), participants were asked to correct their hand movement in the direction opposite to the target-jump direction and to bring the hand cursor to a small gray marker placed on the opposite side to the jumped target as depicted in the second row panels of *Figure 6A*. Namely, if the target jumped forward, the hand cursor needed to be moved backward, and vice versa. In both types of blocks, participants were required to move the visual cursor to the new target (marker) location within a duration of 400 ms after the target jump. In other trials (20% of all trials), mechanical perturbations were applied in the same manner as Experiment 2.

The session involving Normal/Mirror blocks was designed to compare visuomotor and proprioceptive-motor responses in these different visual feedback (Normal vs Mirror) conditions (*Figure 6B*). In the Mirror block, visual information (i.e., cursor, start position, and target) was displayed as a mirror-reversed image of that in the Normal block, as shown in *Figure 6B*. In both the Normal and Mirror blocks, a forward or backward target jump (±12°, as drawn in the middle and right panels of *Figure 6B*) occurred in pseudo-randomly chosen trials (30% of trials for each jump). The time limit for the adjustment in this session was the same as in the Pro/Anti-reaction session.

**Table 2.** Trial conditions in Experiment 3.

| Target | Cursor | Perturbation | # of trials |
|---|---|---|---|
| Std | Show | MP | 4 |
| | | N | 6 |
| | Short-hide | MP | 4 |
| | | N | 6 |
| | Middle-hide | MP | 4 |
| | | N | 6 |
| | Long-hide | MP | 4 |
| | | N | 6 |
| Near | Show | N | 10 |
| | Long-hide | N | 10 |
| Far | Show | N | 10 |
| | Long-hide | N | 10 |

The mechanical perturbation was applied in pseudo-randomly selected trials (20%) as in the Pro/Anti-reaction blocks.

One experimental block consisted of 60 trials (12 baselines, 12 mechanical perturbations, 18 forward jumps, and 18 backward jumps, in random order). In each session, participants performed two types of blocks, two times each, and the whole order of these blocks was randomized. The order of sessions was counter-balanced across participants. All conditions are summarized in *Tables 3* and *4*.

## Data collection and analysis

Hand angle was measured with a rotary encoder (resolution of 0.0055°) attached to the manipulandum and sampled at 500 Hz. Its velocity and acceleration were obtained by calculating the differences between the flexion angles at each sampling frame and applying a low-pass filter (fourth order Butterworth filter, 40 Hz cutoff frequency). To evaluate movement accuracy and duration, start and end of hand movements were estimated for each trial. The start was determined as the point where hand velocity exceeded 5% of its peak, and the end was estimated as the point where the hand velocity became less than 5% of its peak and remained below this value for 300 ms. As an index of movement precision, standard deviation of movement endpoints was calculated.

Electromyography (EMG) data were measured from wrist flexor (FCR: Flexor carpi radialis) and extensor muscle (ECR: Extensor carpi radialis) of right hand using surface electrodes (Ag-AgCl disposable electrode, GE Healthcare Japan, Tokyo, Japan). The signals were sampled at 2000 Hz after filtering (0.53–1000 Hz) and amplification (MME-3116, Nihon Kohden, Tokyo, Japan). That data was high-pass filtered (zero-phase lag, fourth order Butterworth filter, with 50 Hz cutoff frequency) to remove motion artifacts and then was rectified. The rectified EMG (rEMG) was aligned on the onset timing of the mechanical perturbation to calculate the inter-trial average. To evaluate the amplitude of stretch reflex components, mean activities in constant time windows from the perturbation onset (background activities: from −50 to 0 ms, short-latency: from 30 to 50 ms, long-latency: from 50 to 100 ms) were calculated with reference to previous studies (*Lee and Tatton, 1982*; *Cluff and Scott, 2013*). To quantify inter-subject variation, the amplitude of the rEMG was normalized using reference activities during isometric contraction against reference torque of 1 Nm, which was recorded before starting the experiment. For visualization, we applied low-pass filter (zero-phase lag fourth order Butterworth filter with cut-off frequency of 100 Hz) to the rEMG and used it in plotting muscle activities during trials.

To examine the relationship between stretch reflex and movement variability, we evaluated the correlation coefficient between them across conditions. For each condition, we calculated the mean amplitude of the long-latency stretch reflex and the standard deviation of endpoints in unperturbed trials. Then, the correlation coefficient of these values across conditions was computed for each participant. We evaluated the statistical significance of the group mean of the correlation coefficients by a confidence interval with bootstrap resampling method.

In Experiment 4, we evaluated the quick motor responses evoked by target jumps as well as stretch reflexes. Angular accelerations of hand motion were temporally aligned using the jump onset of the visual target. To calculate response latency from the target jump, one-tailed successive *t*-tests (*Crevecoeur et al., 2016*; *Diamond et al., 2015*; *Prablanc and Martin, 1992*) were performed between the two acceleration data sets of the forward jump and for the backward jump for each participant. We defined the response onset as the first time at which the p-values of the *t*-tests were lower than 0.05 at 10 consecutive samples. We also calculated the response latency from EMG signal of both flexor and extensor muscles. The rEMG was low-pass filtered (100 Hz cutoff frequency, zero-

**Table 3.** Session and bock conditions in Experiment 4.

| Session | Block (randomized in each session) |
|---|---|
| Pro/Anti | Pro-reaction x 2 |
| | Anti-reaction x 2 |
| Normal/Mirror | Normal x 2 |
| | Mirror x 2 |

**Table 4.** Trial conditions in each block in Experiment 4.

| MP/Target jump | # of trials (randomized in each block) |
|---|---|
| No (baseline) | 12 |
| MP | 12 |
| Forward jump | 18 |
| Backward jump | 18 |

phase lag, fourth order Butterworth filter) to obtain signal envelope, and then its response onset was calculated in a similar manner (but 50 consecutive samples) with that of acceleration.

To characterize the changes of the reflexive visuomotor responses, we evaluated the early component of muscle response elicited by the target jump. Previous studies (*Day and Lyon, 2000*; *Franklin and Wolpert, 2008*) showed that participants exhibit reflexive response toward the target jump direction in the Pro task, but cannot reverse it from the target jump direction in the Anti task. Therefore, in this study, the reflexive component (v-EMG) was calculated by taking average of rEMG during the interval between the response onsets in the Pro and Anti tasks.

In Experiment 1, a subset of participants showed a tendency to change their muscle activity due to co-contraction resulting from distortions of visual feedback. Since stretch reflexes induced by mechanical perturbation are known to scale with background muscle activity (*Bedingham and Tatton, 1984*), background changes caused by co-contraction could affect the amplitude of the stretch reflex. To rule out this possibility, participants were excluded from analysis (n = 7) if their mean background activity in any visual rotation condition was outside of ±25% range of the total average across conditions.

To remove outliers, we discarded trials from the analysis if the time average of the background muscle activity, the long-latency stretch reflex, or the reflexive visuomotor responses deviated by more than two standard deviations from the median of each condition. In addition, unperturbed trials were also discarded if movement duration or endpoint position was more than three times the median absolute deviation away from the median of each condition. In total, 2.2% trials were excluded, and the most trials removed from an individual across all experiments was 4.9%.

## Experimental design and statistical analysis

All of the experiments and analyses were conducted using a within-subject design. We conducted repeated measures ANOVA to test statistical differences of behavioral data among conditions. Results of Experiment 1 were analyzed by one-way ANOVA with a factor of rotation angle of visual feedback followed by Tukey-HSD test as post-hoc analysis. Results of Experiment 2 were analyzed by two-way repeated measures ANOVA with factors of cursor direction (Normal or Mirror) and cursor visibility (Show or Hide). Results of Experiment 3 were analyzed by one-way ANOVA with a factor of locations of eliminating the visual feedback followed by Tukey-HSD test as post-hoc analysis. Results of Experiment 4 were analyzed by paired *t*-test to see significant differences between trial types for each session. The number of participants (sample size) was determined by referring to previous studies of stretch reflex modulation (*Kimura et al., 2006*; *Kurtzer et al., 2008*). To examine the short latency reflex, a larger number of participants were recruited by referring to a previous study (*Weiler et al., 2019*).

## Acknowledgements

This work was supported by Grants-in-Aid for Scientific Research (JP16H06566) from Japan Society for the Promotion of Science to HG. We thank J De Havas for comments on the manuscript.

## Additional information

### Competing interests

Sho Ito: Sho Ito is affiliated with Nippon Telegraph and Telephone Co. The author has no other competing interests to declare. Hiroaki Gomi: Hiroaki Gomi is affiliated with Nippon Telegraph and Telephone Co. The author has no other competing interests to declare.

### Funding

| Funder | Grant reference number | Author |
| --- | --- | --- |
| Japan Society for the Promotion of Science | JP16H06566 | Hiroaki Gomi |

The funders had no role in study design, data collection and interpretation, or the decision to submit the work for publication.

### Author contributions

Sho Ito, Data curation, Software, Formal analysis, Validation, Investigation, Methodology; Hiroaki Gomi, Conceptualization, Resources, Software, Supervision, Funding acquisition, Validation, Investigation, Methodology, Project administration

### Author ORCIDs

Hiroaki Gomi https://orcid.org/0000-0003-3541-2251

### Ethics

Human subjects: All gave written informed consent to participate in the experiments. All of the experimental protocols were approved (H28-011, H31-007) by NTT Communication Science Laboratories Ethics Committee.

### Decision letter and Author response

Decision letter https://doi.org/10.7554/eLife.52380.sa1
Author response https://doi.org/10.7554/eLife.52380.sa2

## Additional files

### Supplementary files

- Source code 1. Individual EMG and kinematic data.

- Transparent reporting form

### Data availability

All data generated or analysed during this study are included in the manuscript. Source data files have been provided for Figures 2, 3, 4, 5, and 8 for all experiments.

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
