## [Decision Letter]

**Acceptance summary:**

In general, the manuscript includes four experiments that address the influence of visual vs. proprioceptive feedback on stretch reflexes. Transforming or eliminating visual stimuli modulates the stretch reflex, even if the wrist's movement is relatively constant. These novel findings suggest that both visual and proprioceptive factors can influence the stretch reflex.

**Decision letter after peer review:**

Thank you for submitting your article "Visually-updated hand state estimates modulate the proprioceptive reflex independently of motor task requirements" for consideration by *eLife*. Your article has been reviewed by three peer reviewers, including Erin Cressman as the Reviewing Editor and Reviewer #2, and the evaluation has been overseen by Joshua Gold as the Senior Editor.

The reviewers have discussed the reviews with one another and the Reviewing Editor has drafted this decision to help you prepare a revised submission.

Summary:

In the article, "Visually-updated hand state estimates modulate the proprioceptive reflex independently of motor task requirements," the authors conduct a series of four experiments to determine whether state estimates contributing to stretch reflexes are represented solely by proprioceptive information or by multimodal information. In the first three experiments, the authors translated or eliminated visual information. In general, they found that the stretch reflex (i.e., gain of proprioceptive feedback) was attenuated with increasing visuomotor manipulations or elimination durations. In the fourth experiment, the visual-motor mapping was also manipulated (i.e., participants had to Pro- or Anti-point). The stretch reflex was not attenuated under different visual-motor mapping conditions. The authors put forward that these results argue against a general inhibition of quick sensorimotor control and instead indicate that attenuation of the stretch reflex is due to an increase in the uncertainty of estimating hand states, implicating multimodal contributions to the generation of stretch reflexes.

Overall, this manuscript is well written, interesting, and timely. The reviewers did note that there were no issues with respect to the technical aspects of the paper. We have the following suggestions to improve the manuscript.

Essential revisions:

1) All three reviewers commented that the evidence to support the claim that changes in the long latency component of the stretch reflex are due to increased uncertainty in estimating hand states is weak. To begin, the introduction of a visual-motor misalignment across the experiments does not create a visual distortion per se (i.e., the visual signal is not degraded somehow or made less reliable). Thus, the visual-motor misalignment would not be expected to affect visual variance/precision. Rather, it may affect visual accuracy/bias (or non-specific responses), which has not been found to affect modality weightings in multisensory integration.

In accordance with the point raised above, in certain experiments (e.g., Experiment 2), there is not a clear trend between variability in movements and flexor EMG. If changes in the certainty of state estimation drive attenuation of the stretch reflex as suggested by the authors, then the stretch-reflex should be lower in the hide-normal condition compared to the show-mirror condition. Can the authors show this negative relationship between endpoint variability and reflex amplitude across participant data that they elude to (what do correlational values look like)?

Given that variable errors are only reported for some of the conditions, and systematic errors are not reported for any of the conditions, it is difficult to assess whether any of the experimental manipulations affected performance on the motor task. This is important because if changes in the visual display cause changes in the movement, then we can't conclude if the stretch reflex modulation comes from the visual stimuli changing or from the movement changing. Analysis of the movements with regards to accuracy and movement time are required across all four experiments, and their relationship to the flexor EMG.

Moreover, the authors should consider the alternate viewpoint. How does visual feedback improve the state estimate? If the gain is higher with visual and proprioceptive feedbacks, it implies that the integration is performed by taking different delays into account, otherwise fusing perturbation-related motion with visual information from before the perturbation should elicit a reduction in response gains in the combined case, because during a brief interval the perturbation is not yet seen. Thus, the integration of visual and proprioceptive feedback produces an increase in gain (assuming that the distortion alters the reliability) only if the integration is performed by taking dynamics and delays into account. In that regard and contrary to the authors, the reviewers suggest that the current results stand in good agreement with those reported in Crevecoeur et al., (2016). The reviewers recommend that the assumption between distortion and reliability needs to be explored in greater detail and explicitly presented, and that efforts be made to better justify or show in theory why or under which circumstances a reduction in response gain was expected under altered visual condition.

2) A second point raised by the reviewers was the low sample sizes across all experiments. There seems to be a trend in the short latency component of the stretch reflex (Figure 2C), which must be verified with larger sample sizes. If confirmed, this trend would have a clear impact on the interpretation. based on solid theoretical grounds. For example, consider results from Yang et al., 2011 who showed that it takes at least 70ms, and other studies with visuomotor perturbations found values closer to 100ms, to integrate new visual information into a goal-directed response (e.g., Pisella, 2000; Cressman et al., 2006). It seems reasonable to consider that upon changes in visual feedback, participants changed their control policy by reducing the response gain in a non-specific manner. This can be checked in part by looking at short latency with more subjects. The authors do demonstrate that there can be differences in visuomotor and proprioceptive systems, but it does not rule out a non-specific reduction in each modality. The underlying circuits are different, and it would be surprising to see exactly the same reduction in both.

3) A third revision required is the analysis of the "catch trials" to ensure that subjects were consistently attending to the visual feedback. The reviewers can see the value of keeping the required hand movement constant across conditions, so that only the visual display changes. However, as mentioned above, we don't know if the actual movements were in fact constant across conditions. Moreover, repeating the same movement many times gives subjects the opportunity to memorize the motion and learn to execute it without paying attention to the visual feedback. Experiment 1 makes it appear that some trials contained a visual perturbation to prevent this, but no analysis of those trials is presented to confirm that subjects were consistently attending to the visual feedback.

4) All reviewers had difficulty following the experimental manipulations and hence trials included in some of the experiments (specifically, with regards to Experiment #4). Please clarify the different types of trials and instructions for all experiments (e.g., what did participants have to do in the Anti-reaction condition?).

5) With regards to statistics, it seems that:i) Statistics regarding determination of hand correction time in Experiment 4 are not clear. Is response latency evaluated based on a mean path/performance, individual trials per participant, or all trials? Given that a large number of one-tailed *t*-tests were performed, what adjustments were made for the multiple comparisons? And why was this method necessary? Why not simply define a threshold value of acceleration?ii) Why analyze each group in Experiment 1 with separate one-way ANOVAs? A mixed-model ANOVA with factors "group" (between-subjects) and "rotation angle" (within-subject) would be more straightforward.

---

## [Author Response]

Essential revisions:1) All three reviewers commented that the evidence to support the claim that changes in the long latency component of the stretch reflex are due to increased uncertainty in estimating hand states is weak. To begin, the introduction of a visual-motor misalignment across the experiments does not create a visual distortion per se (i.e., the visual signal is not degraded somehow or made less reliable). Thus, the visual-motor misalignment would not be expected to affect visual variance/precision. Rather, it may affect visual accuracy/bias (or non-specific responses), which has not been found to affect modality weightings in multisensory integration.

Thank you for giving an important comment. As in the above comment, visual rotation apparently does not give any ‘visual’ distortion but gives distortion of the visuomotor map by visual feedback. The terms “distortion of visual feedback”, “visual feedback distortion” and “distortion of visuomotor map” are frequently used in the studies of motor learning with visuomotor rotation (over 100 times by Google scholar). Therefore, we would like to keep using the term “distortion of (limb) visual feedback” and “visual feedback distortion” in our revised manuscript. Note that the term “visual distortion” used two times in the original manuscript has been changed to “visual feedback distortion” in the revised manuscript.

But, as in the above comment, many previous studies of visuomotor learning, including our previous studies, frequently used visual rotation paradigms to mainly focus on the shift of the endpoint bias (e.g., Mazzoni and Krakauer, 2006; Saijo and Gomi, 2010, 2012). In contrast to these studies of two-dimensional movements, the current study used one degree-of-freedom movement (wrist rotation only) and showed rotated visual feedback. We did not observe any biases of the endpoints in all rotated conditions (45°, 90°, 135°, 180°) from the baseline condition (0°) in Experiment 1. This is likely because tasks involving one dimensional movement are relatively simple, compared to those using two dimensional movements. However, we found in our current study that the variances (SD) of the endpoint for the 135° and 180° conditions were significantly greater than that for the 0° condition, suggesting that the large rotational distortion does increase the uncertainty of the states used for the motor control.

When considering movements with altered visual feedback more generally, a large angle of visual rotation, as well as the mirror condition, has been shown to increase the reaction time for the movements in question (Saijo and Gomi, 2010; Fernandez-Ruiz et al., 2011; Telgen et al., 2014). This means that these visuomotor transformations are too complex to easily reconcile the mismatch between the visual and proprioceptive information. As a result of this complexity, movement trajectories become variant by visual distortion even after some amount of learning as shown in (Buch et al., 2003). We can therefore imagine that these transformations could deteriorate the integration of the visual and proprioceptive signals to represent the hand states, resulting in large variances of the endpoints for the large visual rotation conditions.

In the revised manuscript, we have added an analysis of endpoint variances in all the rotational conditions of Experiment 1, after increasing the number of participants in the Agonist group in response to another comment from the editor/reviewer (see also our response below). We should note that the endpoint detection method has been improved to stably detect the end of movement. Corresponding sentences in Materials and methods section have been changed. We also added the above examinations to subsection “Visual stimulus effects on motor correction”.

In accordance with the point raised above, in certain experiments (e.g., Experiment 2), there is not a clear trend between variability in movements and flexor EMG. If changes in the certainty of state estimation drive attenuation of the stretch reflex as suggested by the authors, then the stretch-reflex should be lower in the hide-normal condition compared to the show-mirror condition. Can the authors show this negative relationship between endpoint variability and reflex amplitude across participant data that they elude to (what do correlational values look like)?

We greatly appreciate this thoughtful examination of our data. Accordingly, we calculated the correlation values between the endpoint variability and reflex amplitude for each participant in Experiment 2. The mean of the correlation coefficient was -0.73 (bootstrap 95% CI [-0.79 -0.66]), which was significantly different from the null. Although this suggests a statistically significant relationship between endpoint variance and reflex change, the correlation values were not high. If we assume that endpoint variance was perfectly correlated with the representation of state uncertainty, we may be able to expect higher correlation between the changes of reflex and endpoint variance. However, since the endpoint variance could be also affected (reduced) by an online visual correction in the visual feedback condition, the endpoint variances in Normal-Show and Mirror-Show conditions would tend to be underestimated. Namely, differences in the SDs of endpoints between the Hide and Show conditions could be overestimated as an index of uncertainty of state representation. This confounding factor could deteriorate the symmetrical relationship between the endpoint SD change (Figure 4C) and reflex amplitude change (Figures 4E).

Negative correlation between the changes in reflex and in endpoint SD was consistently found in Experiment 1 and Experiment 3, strongly supporting the idea of stretch reflex modulation by the uncertainty of estimated states. In the revised manuscript, the correlation values of Experiment 1, Experiment 2, and Experiment 3 have been described in Results section.

Given that variable errors are only reported for some of the conditions, and systematic errors are not reported for any of the conditions, it is difficult to assess whether any of the experimental manipulations affected performance on the motor task. This is important because if changes in the visual display cause changes in the movement, then we can't conclude if the stretch reflex modulation comes from the visual stimuli changing or from the movement changing. Analysis of the movements with regards to accuracy and movement time are required across all four experiments, and their relationship to the flexor EMG.

Thank you for your careful consideration and apologies for the insufficient information in the original manuscript. In the revised manuscript, movement duration, endpoint position (biases), endpoint standard deviation (variance), and peak velocity of the movements in all the experiments have been reported (figure numbers in the revised manuscript are listed below). As a result, we found significant increases in endpoint variance not only in Experiment 2 but also in all other experiments. If we can assume that the endpoint variance is a reasonable marker of the uncertainty of estimated states as examined in previous study (Wolpert et al., 1995; Izawa and Shadmehr, 2008), this result strengthens the idea that the stretch reflex gain is regulated by the uncertainty of state representation.

As for the other kinematic variables, in Experiment 1, endpoints in all the rotated conditions did not differ from that in the baseline condition (0°) although we found significant difference in endpoint position between 45° and 90° in the post-hoc comparison, as described in Figure 2—figure supplement 1B. The difference in mean endpoint, therefore, does not account for the stretch reflex reductions in larger visual rotation angles. In addition, other indexes (movement duration, peak velocity, and background EMG) were not significantly different across cursor rotation angles. Thus, the stretch reflex modulation is not attributable to changes in movement kinematics.

In Experiment 2 and Experiment 4, although peak velocities were not different among conditions, small but significant differences in endpoint position and in movement duration were found, probably due to the cursor elimination and mirror-reversal. Therefore, to justify the comparison of the stretch reflex responses in different visual feedback conditions, hand velocity just before the application of mechanical perturbation, and background EMG (BGA) are also reported in figure supplements (Figure 4—figure supplement 1D and 1E and Figure 8—figure supplement 1E and 1F, respectively) in the revised manuscript. There was no difference in those values among the conditions in each experiment, suggesting the difference of stretch reflex modulation could not be attributed to changes in movement kinematics and background muscle activity.

For Experiment 1 (Agonist group)

Movement duration: Figure 2—figure supplement 1A

Endpoint (bias): Figure 2—figure supplement 1B

Endpoint SD: Figure 2B

Peak velocity: Figure 2—figure supplement 1C

For Experiment 1 (Antagonist group)

Movement duration: Figure 3—figure supplement 1A

Endpoint (bias): Figure 3—figure supplement 1B

Endpoint SD: Figure 3B

Peak velocity: Figure 3—figure supplement 1C

For Experiment 2

Movement duration: Figure 4—figure supplement 1A

Endpoint (bias): Figure 4—figure supplement 1B

Endpoint SD: Figure 4C

Peak velocity: Figure 4—figure supplement 1C

Hand velocity at MP onset: Figure 4—figure supplement 1D

Flexor background EMG (BGA): Figure 4—figure supplement 1E

For Experiment 3

Movement duration: Figure 5—figure supplement 1A,

Endpoint (bias): Figure 5—figure supplement 1B,

Endpoint SD: Figure 5C,

Peak velocity: Figure 5—figure supplement 1C,

Flexor background EMG (BGA): Figure 5—figure supplement 1D,

For Experiment 4

Movement duration: Figure 8—figure supplement 1A

Endpoint (bias): Figure 8—figure supplement 1B

Endpoint SD: Figure 8—figure supplement 1C

Peak velocity: Figure 8—figure supplement 1D

Hand velocity at MP onset: Figure 8—figure supplement 1E

Flexor background EMG (BGA): Figure 8—figure supplement 1F

Moreover, the authors should consider the alternate viewpoint. How does visual feedback improve the state estimate? If the gain is higher with visual and proprioceptive feedbacks, it implies that the integration is performed by taking different delays into account, otherwise fusing perturbation-related motion with visual information from before the perturbation should elicit a reduction in response gains in the combined case, because during a brief interval the perturbation is not yet seen. Thus, the integration of visual and proprioceptive feedback produces an increase in gain (assuming that the distortion alters the reliability) only if the integration is performed by taking dynamics and delays into account. In that regard and contrary to the authors, the reviewers suggest that the current results stand in good agreement with those reported in Crevecoeur et al., (2016). The reviewers recommend that the assumption between distortion and reliability needs to be explored in greater detail and explicitly presented, and that efforts be made to better justify or show in theory why or under which circumstances a reduction in response gain was expected under altered visual condition.

Thank you for insightful comment. As in the comment, we completely agree that the different delays should be taken into account for sensory integration, which greatly affects the state representation. Our hypothesis is not contradictory to the Crevecoeur et al., (2016), which showed the importance of the short delay of proprioceptive information in estimating limb states. However, only with that formulation, we think that the reduction of reflex gain observed in our experiments cannot be explained. If we use an optimal control model for the visuomotor control of reaching, some extension of the model would be needed to reproduce our observation. There might be at least two possible ways to explain the gain reduction of stretch reflex.

One possibility is that the optimal control policy calculating feedback gain is modified to evaluate the uncertainty of the state representation. Increase in uncertainty of state estimates can cause reduction of the proprioceptive feedback gain if smaller gain is ‘optimal’ by evaluating uncertainty in the policy. In order to avoid an unnecessary or harmful compensatory motor response, the policy (or strategy) change of stretch reflex reduction would be reasonable when proprioceptive information itself is insufficient to represent the reliable limb state in the task (visual) space. In such a case, the short latency stretch reflex could be also changed by the top-down regulation, even if it is not involved in high level state estimation.

Another possibility is that the uncertainty increase in estimated states induces a reduction of the filter gain in state estimation process, which could lead to a tracking delay in state estimation, resulting in a reduction of the corrective command for a transient perturbation. As described in the manuscript, this possibility of gain reduction due to an increase in state uncertainty was examined previously for visuomotor responses (Izawa and Shadmehr, 2008). By considering the influence of state uncertainty on regulating proprioceptive-motor response shown in our study, it seems possible in future study to extend the computational model to include multimodal integration for reaching movements. In both possibilities, we need to assume that at least a part of motor command is generated in a feedforward fashion to realize the predefined speed of movement. This idea has been supported by many experimental and model studies (Wolpert et al., 1998; Kawato, 1999; Sabes, 2000; Bastian, 2006; Saijo and Gomi, 2010).

The above discussion has been added in the Discussion section (Uncertainty in state estimates affects feedback gain) in the revised manuscript.

2) A second point raised by the reviewers was the low sample sizes across all experiments. There seems to be a trend in the short latency component of the stretch reflex (Figure 2C), which must be verified with larger sample sizes. If confirmed, this trend would have a clear impact on the interpretation. based on solid theoretical grounds. For example, consider results from Yang et al., 2011 who showed that it takes at least 70ms, and other studies with visuomotor perturbations found values closer to 100ms, to integrate new visual information into a goal-directed response (e.g., Pisella, 2000; Cressman et al., 2006). It seems reasonable to consider that upon changes in visual feedback, participants changed their control policy by reducing the response gain in a non-specific manner. This can be checked in part by looking at short latency with more subjects. The authors do demonstrate that there can be differences in visuomotor and proprioceptive systems, but it does not rule out a non-specific reduction in each modality. The underlying circuits are different, and it would be surprising to see exactly the same reduction in both.

According to the above request, we have added eight participants in the Agonist group of Experiment 1. As a result, we found significant differences of short-latency reflex components in the 90° and 135° conditions from the baseline (0°) but did not find the difference in the 180° condition from the baseline. This modulation trend of short-latency stretch reflex was not comparable to that of long-latency stretch reflex as shown in Figure 2D. Additionally, we did not find any modulation of the short-latency stretch reflex component in Experiment 1 Antagonist group, Experiment 2 (Normal/Mirror, Show/Hide), Experiment 3 (Hide duration change), or Normal/Mirror conditions in Experiment 4, while the modulation of long-latency reflex was consistently observed in those experiments. Since we mainly focus on the modulation of long-latency components in this study and we could not find any systematic modulation of the short-latency components in Experiment 1, we did not add participants in the other experiments.

Of course, we do not deny the possibility of the modulation of short-latency component by visual information. In addition to the relatively quick effect of visual stimuli on the motor responses (Cressman et al., 2006; Pisella et al., 2000; Yang et al., 2011), previous study clearly exhibited the short-latency component modulation by visual stimuli (Weiler et al., 2019). The short-latency component modulation observed in our study could be partly explained by a policy change of proprioceptive specific gain reduction as examined in the previous response. As previously shown (Weiler et al., 2019), the short-latency stretch reflex could be modulated in task-specific manner.

However, as explained above, we did not observe similar modulations of the short-latency and long-latency components. Rather, we observed a significant modulation of endpoint variabilities by visual distortions (rotation and mirror) and by cursor elimination, and that the amplitudes of long-latency components were negatively correlated with the observed endpoint variabilities. Based on the recent considerations of multimodal state estimation and optimal feedback models, we suggest a possibility that the stretch reflex modulation is ascribed to the uncertainty of estimated states. Some class of reduction of stretch reflex in a non-specific (i.e., no task-related) manner may be explained by this hidden linkage between state uncertainty and reflex modulation. The above examination has been added in Discussion (Subsections of ‘Visual stimulus effects on motor correction’ and ‘Uncertainty in state estimates affects feedback gain’).

3) A third revision required is the analysis of the "catch trials" to ensure that subjects were consistently attending to the visual feedback. The reviewers can see the value of keeping the required hand movement constant across conditions, so that only the visual display changes. However, as mentioned above, we don't know if the actual movements were in fact constant across conditions. Moreover, repeating the same movement many times gives subjects the opportunity to memorize the motion and learn to execute it without paying attention to the visual feedback. Experiment 1 makes it appear that some trials contained a visual perturbation to prevent this, but no analysis of those trials is presented to confirm that subjects were consistently attending to the visual feedback.

Thank you for your careful attention to experimental control. We have analyzed the data of catch trials. Mean endpoints of catch trials of each participant in each rotation angle condition and the corresponding reaction times are shown in Figure 2—figure supplement 1D in the revised manuscript. As shown in the figure, participants correctly shifted movement endpoints in catch trials and those reaction times were sufficiently short, suggesting all participants were consistently attending to the visual cursor feedback. Brief explanation has been added in Results section – in the revised manuscript.

4) All reviewers had difficulty following the experimental manipulations and hence trials included in some of the experiments (specifically, with regards to Experiment #4). Please clarify the different types of trials and instructions for all experiments (e.g., what did participants have to do in the Anti-reaction condition?).

We have added session/block/trial condition summary tables (Table 2, Table 3, and Table 4) for Experiment 3 and Experiment 4 to clarify the experiments in the revised manuscript, as written for Experiment 2. Additionally, method descriptions of Experiment 3 and Experiment 4 have been totally revised to clarify the trial conditions and tasks. Since the trial conditions of Experiment 1 were so simple, we just left the explanation of the experimental manipulation in the main text.

5) With regards to statistics, it seems that:i. Statistics regarding determination of hand correction time in Experiment 4 are not clear. Is response latency evaluated based on a mean path/performance, individual trials per participant, or all trials? Given that a large number of one-tailed *t*-tests were performed, what adjustments were made for the multiple comparisons? And why was this method necessary? Why not simply define a threshold value of acceleration?

Response latency was estimated by successively comparing two data sets for forward jump and for backward jump in each participant. This method is standardly used to find the divergence points for time series data statistically diverge (Prablanc and Martin, 1992; Diamond et al., 2015; Crevecoeur et al., 2016) without adjustment of statistical thresholds (possibly, because the main purpose of the analysis is not strict p-values in this case). Though onset detection with a threshold value is also commonly used (Oostwoud Wijdenes et al., 2014), estimated latency would be relatively sensitive to the threshold value. Particularly, if response is not much greater than baseline variability, it could be difficult to determine reasonable threshold value for accurate and stable start detection because small threshold value could incorrectly detect baseline change, as latency and large threshold values could estimate latency longer than ‘true’ value. In the current study, we found it difficult to select good and consistent threshold value across participants due to large trial-by-trial variability and noises in acceleration and EMG signals. Thus, we employed successive *t*-tests as a latency detection method. The papers which used similar method have been cited and the explanation of the latency detection has been revised. [Subsection “Data collection and analysis” in the revised manuscript].

ii. Why analyze each group in Experiment 1 with separate one-way ANOVAs? A mixed-model ANOVA with factors "group" (between-subjects) and "rotation angle" (within-subject) would be more straightforward.

In Experiment 1, we are mainly interested in the effects of visual feedback on the stretch reflexes of flexor (agonist) muscle in one group and of extensor (antagonist) muscle in the other group, separately. Two experiments were done independently, and mechanical perturbations were applied in opposite directions. Therefore, even though basic tasks were the same in these experiments, we think that it is unnecessary (or inappropriate) to compare the data between these conditions. Additionally, because baseline activities of agonist and antagonist muscles are quite different and independent in nature, we cannot assume homogeneity of variance between these two variables. Thus, we think it inappropriate to directly compare these variables in a single ANOVA.